**The eFLaG dataset: development and evaluation of nationally consistent projections of future flows and groundwater based on UKCP18**

Jamie Hannaford[1,2], Jonathan D. Mackay[3,4], Matthew Ascott[5], Victoria A. Bell[1], Thomas Chitson[1], Steven Cole[1], Christian Counsell[6], Mason Durant[6], Christopher R. Jackson[3], Alison L. Kay[1], Rosanna A. Lane[1], Majdi Mansour[3], Robert Moore[1], Simon Parry[1], Alison C. Rudd [1], Michael Simpson[6], Katie Facer-Childs[1], Stephen Turner[1], John R. Wallbank[1], Steven Wells[1], Amy Wilcox[6]

[1]UK Centre for Ecology & Hydrology, Maclean Building, Benson Lane, Crowmarsh Gifford, Wallingford, Oxon, OX10 8BB, UK

[2]Irish Climate Analysis and Research UnitS (ICARUS), Maynooth University, Ireland

[3]British Geological Survey, Keyworth, Nottingham, NG12 5GG, UK

[4]School of Geography, Earth and Environmental Sciences, University of Birmingham, Edgbaston, B15 2TT, UK

[5]British Geological Survey, Maclean Building, Benson Lane, Crowmarsh Gifford, Wallingford, Oxon, OX10 8BB, UK

[6]HR Wallingford, Howbery Park, Crowmarsh Gifford, OX10 8BA, UK

Corresponding authors:

Jamie Hannaford jaha@ceh.ac.uk

Jonathan MacKay joncka@bgs.ac.uk

## Abstract

This paper details the development and evaluation of an 'enhanced future FLows and Groundwater' (eFLaG) dataset of nationally consistent hydrological projections for the UK, based on the latest UK Climate Projections (UKCP18). The hydrological projections are derived from a range of river flow models (Grid-to-Grid, PDM, GR4J and GR6J), to provide an indication of hydrological model uncertainty, as well as groundwater level (Aquimod) and groundwater recharge (ZOODRM) models. A 12-member ensemble of transient projections of present and future (up to 2080) daily river flows, groundwater levels and groundwater recharge were produced using bias corrected data from the UKCP18 Regional (12km) climate ensemble. Projections are provided for 200 river catchments, 54 groundwater level boreholes and 558 groundwater bodies, all sampling across the diverse hydrological and geological conditions of the UK. An evaluation was carried out, to appraise the quality of hydrological model simulations against observations and also to appraise the reliability of hydrological models driven by the RCM ensemble, in terms of their capacity to reproduce hydrological regimes in the current period. The dataset was originally conceived as a prototype climate service for drought planning for the UK water sector, so has been developed with drought, low river flow and low groundwater level applications as the primary focus. The evaluation metrics show that river flows and groundwater levels are, for the majority of catchments and boreholes, well simulated across the flow and level regime, meaning that the eFLaG dataset could be applied to a wider range of water resources research and management contexts, pending a full evaluation for the designated purpose. Only a single climate model and emissions scenario are used, so any applications should ideally contextualise the outcomes with other climate model/scenario combinations.

56

## 1. Introduction

58

This paper details the development and evaluation of an 'enhanced future FLows and Groundwater' (hereafter referred to as "eFLaG") dataset of nationally consistent, and spatially coherent, hydrological (river flow and groundwater) projections for the UK, based on UKCP18 – the latest climate projections for the UK from the UK Climate Projections programme (Murphy et al. 2018). eFLaG provides a successor to the Future Flows and Groundwater Levels (FFGWL) dataset (Prudhomme et al. 2013), which was based on the UKCP09 projections (Murphy et al. 2010).

The eFLaG dataset was developed specifically as a demonstration climate service for use by the water industry for water resources and drought planning, and hence by

design is focused on future projections of drought, low river flows and low groundwater
levels. By providing a consistent dataset of future projections of these variables, eFLaG
can potentially support a wide range of applications across other sectors. The
predecessor, FFGWL, has been widely used within the water industry, but also found
very wide application for diverse research purposes (see Section 8).
As in FFGWL, in eFLaG the climate projections are used as input to a range of
hydrological models to provide nationally consistent, spatially coherent projections of
river flow and groundwater levels for the 21$^{st}$ century. The use of an ensemble of river
flow models also provides information on hydrological model uncertainty. As well as
using an updated set of climate projections, eFLaG capitalises on advances in
national-scale river flow and groundwater modelling since FFGWL, and detailed
evaluation of the applicability of models for drought simulation, notably research under
the NERC Drought and Water Scarcity (DWS) Programme (e.g. Rudd et al. 2017;
Smith et al. 2019).

## Previous research on hydrological projections

There is a long history of climate change impact assessment within the UK water
industry and academia, which we do not review in detail here. Watts et al. (2015)
provides an overview of past research (up to around 2013) on climate projections
relevant for the water sector, including for future water resources and drought. More
recently, Chan et al. (2022) provide an in-depth review on the evolution of the use of
climate change projections for hydrological applications. Here, we briefly address
some pertinent developments in river flow projections since FFGWL.
The original FFGWL did not present an assessment of future drought risk, other than
seasonal river flows (Prudhomme et al. 2012) and groundwater levels (Jackson et al.
2015), which suggested: pronounced decreases in future summer flows; reductions in
annual average groundwater levels; and increases (decreases) in winter (summer)
groundwater levels. Since then, the original FFGWL projections have been used in a
number of hydrological impact studies. Collet et al. (2018) presented a probabilistic
appraisal of future river flow drought (and flood) hazard in the UK, showing hydro-
hazard 'hot-spots' in western Britain and northeast Scotland, especially during the
autumn. Hughes et al. (2021) used the ZOODRM distributed groundwater recharge
model to assess changes in 21$^{st}$ century seasonal recharge across river basin districts
and groundwater bodies in the UK based on the FFGWL climate change projections.
The results showed a consistent trend of more recharge being concentrated over fewer
months with increased recharge in winter and decreased recharge in summer.
In addition to UKCP09/FFGWL, other datasets have been developed using different
Global Climate Model (GCM)/Regional Climate Model (RCM)/hydrological modelling

chains. One major development has been the use of large ensemble projections of future climate variables from the Weather@Home RCM (specifically HadRM3P) as part of the MaRIUS project within the DWS Programme (Guillod et al., 2018). The MaRIUS projections provide large ensembles (100+) of past, present (1900–2006) and future (2020–2049 and 2070–2099) climate outputs. These were used as inputs to the national-scale Grid-to-Grid (G2G) hydrological model to provide a similarly large gridded (1km$^2$) dataset of river flow and soil moisture (Bell et al., 2018). Analysis of these datasets has been conducted for drought (Rudd et al. 2019) and low flows (Kay et al. 2018), indicating future increases in hydrological drought severity and spatial extent, and decreases in absolute low flows.

A further source of hydro-meteorological projections now available are those from the EDgE project (End-to-end Demonstrator for improved decision-making for the water sector in Europe), see Samaniego et al. (2019). EDgE delivered an ensemble comprising of two GCMs and four 'impact' models (gridded land surface and hydrological models at a 5x5km scale) for the whole of Europe. Visser-Quinn et al. (2019) analysed future river flow drought risk in this ensemble, using a similar approach to Collet et al. (2018), and found similar results in terms of the spatial distribution and magnitude of future changes in droughts, albeit with some differences arising from the use of different scenarios, GCMs and hydrological models.

While such products may be used for climate adaptation research, the most relevant for eFLaG is the release of UKCP18. To date, relatively few studies using UKCP18 have been published. Kay et al. (2020) made a rapid assessment of UKCP18 impacts on hydrology compared to UKCP09. More recently, Kay (2021), Kay et al. (2021a,b,c), Lane & Kay (2021) and Kay (2022) provided future assessments of potential changes in seasonal mean river flows, high flows and low flows using various UKCP18 products with the G2G hydrological model. They found potential increases in winter mean flows and high flows, and decreases in summer and low flows, albeit with wide uncertainty ranges. In the literature to date, and to the authors' knowledge, there have been no published assessments of future groundwater levels or groundwater recharge using UKCP18 – although groundwater levels driven by UKCP18 are currently being used in the latest operational water resource management plans (e.g. Thames Water, 2023).

In summary, there have been substantial scientific advances in hydrological projections for the UK since Watts et al. (2015) and FFGWL, including some research on future indicators relevant for water resource availability and drought. However, relatively few datasets have been made available to the community since FFGWL. While MaRIUS and EDgE provide complementary hydrological datasets, there remains a need for an accessible dataset based on UKCP18. Existing UKCP18 studies have been focused on time-slice projections and/or used a single hydrological model (e.g. Kay et al., 2021 a,b,c) so there will be significant benefit arising from the eFLaG dataset

of transient projections from a range of hydrological models covering river flows,
groundwater levels and groundwater recharge.


**2.  Outline of dataset and overview of the modelling chain**

In the following sections we set out the methodology behind the eFLaG dataset. This
section firstly provides a brief overview of the various stages of the methodology, and
how our method samples the 'cascade of uncertainty' (Smith et al. 2019) emerging
from the multiplicity of projections and other modelling choices. While the original
FFGWL methodology provided an initial foundation for eFLaG, much has changed in
the decade since that study was commissioned, and the new UKCP18 projections
differ from UKCP09 (e.g. Kay et al. 2020). eFLaG therefore required the development
of a new methodology, which is described in detail in the following sections.
The whole project workflow is illustrated in Fig 1. eFLaG is driven by the UKCP18
dataset, specifically the 'Regional' 12km projections, to which a bias correction is
applied. Section 3 describes the processing of the climate projections, including the
bias correction method. The UKCP18 projections are used as input to three river flow
models (GR, PDM and G2G), one groundwater level model (AquiMod) and one
groundwater recharge model (ZOODRM) to provide simulations for 200 river
catchments, 54 groundwater boreholes and 558 groundwater bodies respectively.
Section 4 provides more detail on how these sites were selected.  Details of the
hydrological models and their calibration are given in Section 5. The evaluation of the
models is covered in sections 6 and 7. Fig 1 also illustrates how all of the eFLaG
projections are feeding into a series of water industry demonstrators, in partnership
with UK water providers (specifically, Dwr Cymru/Welsh Water and Thames Water).
These are not discussed in detail in this paper, but these were relevant for the site
selection and as such are mentioned briefly below.

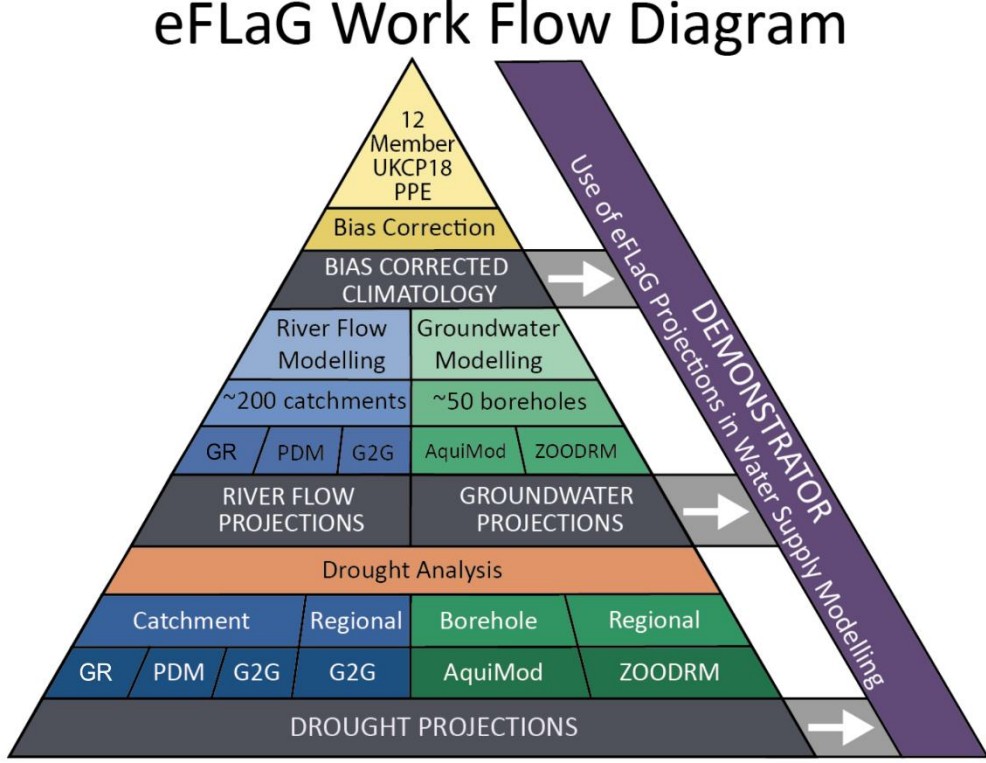

**Figure 1 Project workflow illustrating the stages of analysis described in this paper**

The question of uncertainty in climate impacts modelling is a challenging one that has been explored in a whole range of studies, going back as far as climate projections have been routinely produced from the 1980s. There are inherent uncertainties at every step of the process, from climate emissions scenarios through to climate modelling, and on to environmental modelling (in our case hydrological modelling, which itself has a vast literature when it comes to uncertainty estimation) and then to wider impacts modelling (e.g. in water supply systems). Recently, Smith et al. (2018) presented this issue as a 'cascade of uncertainty' (using widely adopted terminology, e.g. Wilby and Dessai, 2010). Within eFLaG, as with the majority of climate impact applications, it is not possible to sample across all sources of uncertainty. We adopted a pragmatic approach to sampling key sources of uncertainty within the available time and resource constraints. In Table 1, we consider the sources of uncertainty, and our approach to sampling from them. The focus in eFLaG is on uncertainty arising from initial/boundary conditions. Additionally, for the river flow simulations, the uncertainty arising from model choice is also accounted for, embracing models of different type (lumped and distributed) and structure. The effect of different structures of the same model is also included through the use of two versios of one of the models (namely the GR suite).



**Table 1: Sources of uncertainty explored in eFLaG (building on the framework of Smith et al. 2018)**

| Uncertainty Source | Sampling Approach | Details |
|---|---|---|
| **Emissions Scenarios** | One scenario | RCP8.5 |
| **Climate Models** | One model | Hadley Centre GCM |
| **Initial/Boundary Conditions** | 12x member PPE (Perturbed Parameter Ensemble) | PPE perturbs the parameters of the climate model (both the RCM, and the GCM within which it is nested) |
| **Temporal/Spatial Downscaling** | One method | Hadley Centre RCM, monthly mean bias correction |
| **Model Choice** | 3x river flow models 2x groundwater models | GR, PDM, G2G Aquimod, ZOODRM |
| **Model Structure** | 2x model structures for the GR modelling framework | Fixed structure for G2G and PDM, but for GR two different model structures were used (GR4J and GR6J), as discussed in section 4. |
| **Model parameter uncertainty** | Not considered in eFLaG | Not considered in eFLaG |



### 3. UKCP Data Processing


The UKCP18 regional climate projections were created using perturbed-parameter runs of the Hadley Centre global climate model (GCM, HadGEM3-GC3.05) and regional climate model (RCM, HadREM3-GA705) (Murphy et al. 2018). These provide a set of 12 high resolution (12km) spatially consistent climate projections over the UK, covering the period Dec 1980-Nov 2080. The 12-member RCM perturbed parameter ensemble (PPE) is valuable to represent climate model parameter uncertainty; ensemble members are numbered 01–15 excluding 02, 03 and 14 (as there are no RCM equivalents for these GCM PPE members, Murphy et al. 2018 section 4.3), and 01 is the standard parameterisation. However, it is important to note that, as all

ensemble members are based on the same high emissions scenario (RCP8.5) and
underlying climate model structure, they do not represent the full climate uncertainty.
The UKCP18 RCM output was processed to provide the variables needed for
hydrological modelling – namely, 1km gridded and catchment-average time-series of
available precipitation (i.e. after the application of a snow module, see below) and
Potential Evapotranspiration (PET), not itself a UKCP18 output but estimated using
available UKCP18 variables as described below.
The Hadley Centre climate model uses a simplified 360-day year, consisting of twelve
30-day months. The RCM precipitation and temperature time-series are given for this
360-day calendar, and are therefore not consistent with the 365/6-day observed time-
series. Previously, the FFGWL Climate project inserted five (or six in a leap year) days
of zero rainfall into the RCM time-series so that the observed and RCM data were
using comparable calendars (Prudhomme et al., 2012). However, here the data were
kept in the 360-day format, to avoid modifying the time-series with artificial data.

**Precipitation**

Daily precipitation time-series were available for each of the UKCP18 RCM-PPE
members. However, the RCM data showed biases compared to observed precipitation,
as is common for climate data (Murphy et al., 2018; Teutschbein & Seibert, 2012). The
RCM data  substantially over-estimates precipitation for most months (typically by
around 1mm/day for the UK mean, Murphy et al. (2018) Fig 4.4), the exception being
August-October. A simple monthly-mean bias-correction methodology was therefore
applied, through the following steps:
1. The 1km HadUK-Grid observed rainfall product was averaged to 12km for
consistency with the RCM data (Hollis et al., 2019).
2. For each month and grid-cell, change factors were calculated between the RCM
simulated precipitation and observation-based HadUK-Grid time-slice mean of
monthly total rainfall over the period 1981-2010. This resulted in bias-correction
factor grids being made for each month and each RCM ensemble member, as
shown in Fig. 2.
3. The change factor grids were then smoothed to reduce spatial discontinuities,
by updating each grid cell using a weighted combination of the original grid-cell
value and neighbouring values, as in Guillod et al. (2018).
4. To produce bias-corrected precipitation estimates, the RCM simulated
precipitation time-series were multiplied by the bias-correction factor grid for
each month (i.e. all January precipitation was multiplied by the January bias-
correction grids, February precipitation by the February correction grid, etc.).
The bias-corrected precipitation products were then downscaled from 12km to 1km
based on the distribution of the Standard Average Annual Rainfall (SAAR) for the
period 1961-1990, as in previous studies (Bell et al., 2007; Kay & Crooks, 2014). This
involved calculating the ratio of the observed SAAR at 1km to the observed SAAR
averaged up to the 12km RCM grid, and then multiplying RCM precipitation values by
this ratio. This introduces further spatial variability related to typical rainfall patterns,
but the total rainfall across the original 12km RCM grid cell remains unchanged.

**Accounting for snowmelt processes**

A simple snow module was applied to account for snow-melt processes (Bell et al.,
2016). The snow module converted the 1km bias-corrected precipitation into rainfall
plus snowmelt (i.e. available precipitation), based on temperature. This used the
minimum and maximum daily temperatures provided by each RCM ensemble member,
which were first scaled from a 12km resolution to 1km using a lapse rate based on
elevation data. The parameters used in the snow module are given in Supplementary
Info (Table S1).

**Potential evapotranspiration**

Potential evapotranspiration (PET) was not directly available as an RCM output, and
was therefore generated using a range of variables from the RCM-PPE climate time-
series (Table S2). The PET was calculated using the same methodology as the Hydro-
PE dataset (Robinson et al. 2022) except for the use of eFLaG bias-corrected
precipitation data within the interception correction component. This produces
Penman-Monteith PET parameterised for short grass. The equation also included
monthly stomatal resistance values, which were adjusted for the future period to
account for the impact of increased carbon dioxide concentrations on stomata (as in
Rudd & Kay, (2016), based on Kruijt et al., (2008)). The PET data were then copied
down from a 12km to 1km resolution by simply setting all 1km grid cells to the value of
the containing 12km grid cell.

**Outputs**

The 1km gridded time-series of 'available precipitation' and PET were then used to
produce the time-series of catchment-averages required for each of the eFLaG river
catchments and groundwater boreholes. For the river catchments, the catchment
average values were derived using the standard UK National River Flow Archive
approach for catchment average rainfalls, as described in NRFA (2021). For the
boreholes, following Mackay et al. (2014a), averages were taken over the
representative aquifer length which was determined as the groundwater flow path
between the borehole and a single discharge point on a river based on the catchment
geometry and hydrogeology. For the grid-based models, ZOODRM and G2G, the
gridded data were used directly.
The bias-corrected climate outputs are part of the eFLaG dataset described further in
Section 9. For each river catchment and groundwater borehole, bias-corrected data
are available for the observational period, for the purposes of evaluation of the
hydrological model outputs, and for the future. In addition, the gridded bias-corrected
climatology is made available as a separate dataset (Lane and Kay, 2022; see also
the data availability section).

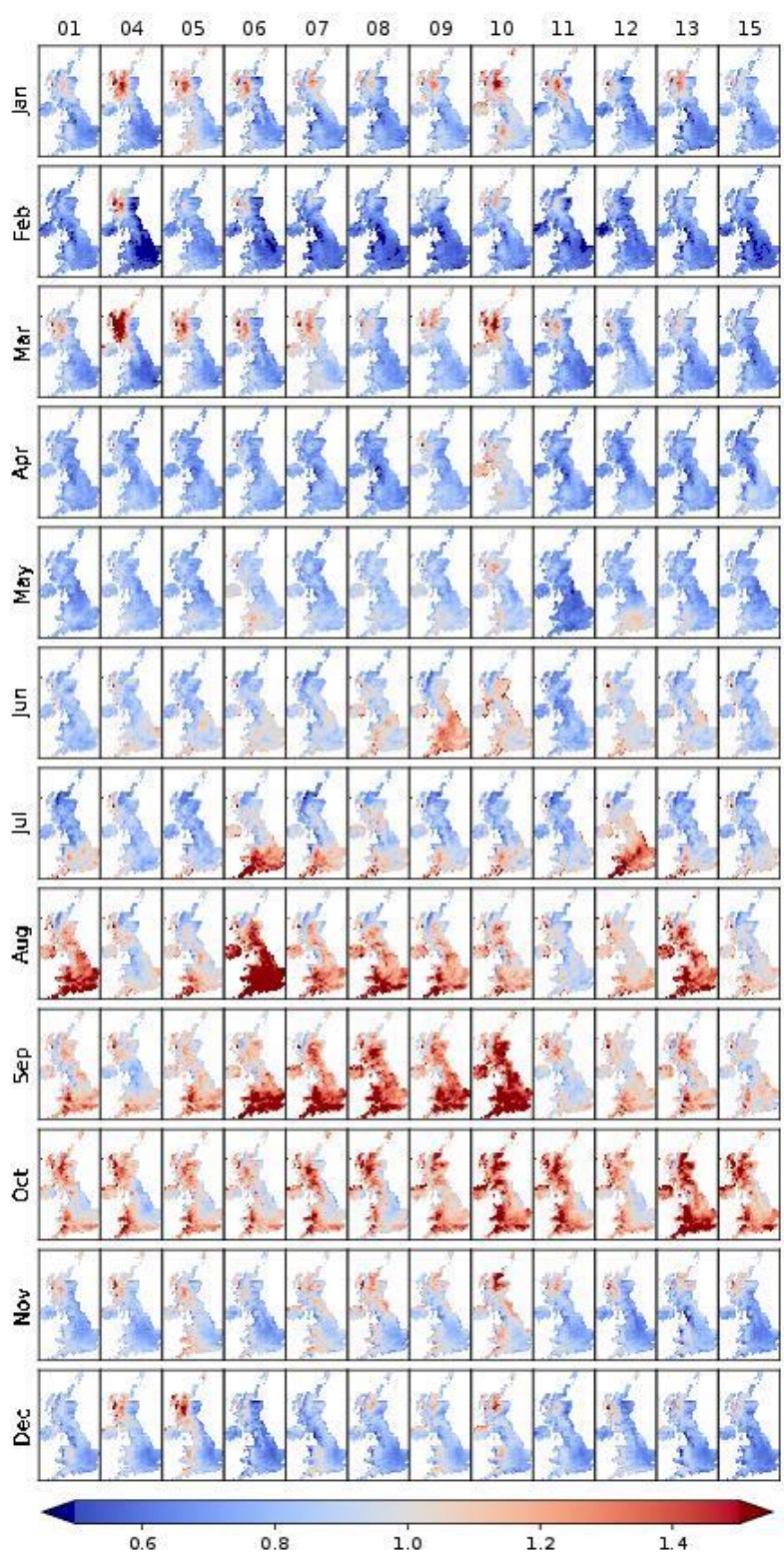



**Figure 2: Bias-correction grids applied to correct monthly precipitation. Values are correction factors used to modify precipitation, with a value of 0.5 halving precipitation, 1 meaning no change to precipitation and 2 doubling precipitation etc. Columns show results from each RCM PPE member, rows show results for each month. Note the column numbers reflect the RCM PPE number (see Sect. 3)**

## 4. Catchment selection

The UK is fortunate to have one of the densest hydrometric networks in the world, with a legacy of strong commitment to data quality and completeness. There are more than 1,500 river flow gauging stations with flow records on the UK National River Flow Archive (NRFA, Dixon et al. 2013 and https://nrfa.ceh.ac.uk/) and more than 180 observation boreholes with groundwater level records on the BGS National Groundwater Level Archive (NGLA). These archives are the principal sources of validated river flow and groundwater level data at the UK scale. A remit of the NRFA and NGLA is to archive data that are useful for a wide variety of applications, primarily focusing on the most strategically important records. However, such catchments are not always the most relevant for the water industry, and water companies often have their own sites on which they undertake analysis. Since the eFLaG project aims to maximise utility for a range of users, the catchment selection strategy considered both research and industry needs.

Detailed site lists and metadata for river flow, groundwater level and groundwater recharge are catalogued on the dataset held on the Environmental Informatics Data Centre (EIDC) (Hannaford et al. 2022).

**River Flows**

To support selection, a metadatabase was assembled for all NRFA gauging stations in the UK, primarily using the NRFA's metadata holdings published on the NRFA website and in the UK Hydrometric Register (Marsh and Hannaford, 2008). Metadata compiled included membership of key national strategic networks (e.g. near-natural Benchmark (UKBN2; Harrigan et al. 2018a) and operational monitoring networks), capitalising on efforts of other projects in quality controlling data and ensuring catchments are fit for purpose. Selection also considered whether catchments were used in previous relevant projects that have simulated river flows for drought analysis. The selection ensured a strong representation of the original FFGWL catchments (with 117 catchments featuring in both) and also overlap with recent modelling endeavours through the DWS Programme (AboutDrought, 2021) projects 'Historic Droughts', 'IMPETUS' and 'MaRIUS' projects, that used several of the models used by eFLaG (specifically G2G, GR4J). In this regard we ensured that 165 eFLaG catchments overlapped with at least one DWS project.

Selection also focused on data quality. Longer record lengths were prioritised and hydrometric quality was evaluated where possible. Given the extent of hydrometric issues (at low flows especially) it is not possible for all sites to have the highest quality data, but where decisions were made on similar sites, quality was considered as a tiebreaker. The selection included 80 Benchmark catchments, but did not seek to focus entirely on natural catchments given the limited range of variability they capture (being

mostly small and clustered in headwaters), and also included large and disturbed sites known to be important for water industry purposes. Artificial influences are prevalent across the UK and have been shown to prominently affect flow regimes (e.g. Rameshwaran et al. 2022) and drought characteristics (Tijdeman et al. 2018) in many catchments. Hence, the incorporation of a range of Benchmark near-natural catchments and artificially influenced sites is important for ensuring representativeness and demonstrating the utility of the different models used, which treat artificial influences differently (Sect 5). Membership of the Benchmark catchments is highlighted in the dataset description, and information on artificial influences can be accessed for all sites on the NRFA website (in station descriptions and 'Factors Affecting Runoff' codes).

Catchment representativeness was also considered, enabling the eFLaG dataset to sample the hydrological variability of the UK. Representativeness was considered by comparing the distribution of eFLaG potential selections relative to various catchment descriptors from the NRFA Hydrometric Register (altitude, area, annual rainfall, Base Flow Index, land cover and so on).

Finally, this activity focused on ensuring water industry relevance. At the national scale, this was achieved by asking stakeholders at an eFLaG workshop for views on additional catchments (Durant et al. 2022). In this way, 12 catchments were added. Similarly, for the regional demonstrators (Dwr Cymru/Welsh Water and Thames Water), water company teams were consulted to gain a better understanding of strategically important flow records for water companies in the case study regions, leading to an additional five catchments.

The final eFLaG dataset consists of 200 catchments (Fig. 3a) giving good geographical coverage and representativeness of the UK.

**Groundwater Levels**

Boreholes were selected to ensure a number of essential criteria were met. Firstly, only those boreholes with the highest-quality records of groundwater level were considered. This required regular (at least monthly) and continuous (at least 10 years in length) records of data from boreholes that are in zones which are not significantly affected by groundwater abstraction.

Secondly, sites were chosen to ensure coverage of the UK's principal aquifers where possible, enabling the eFLaG dataset to sample the hydrogeological variability of the UK. This broadly aligns with the requirements of other national-scale assessments of groundwater resources undertaken as part of the original FFGWL project and the 'Historic Droughts' and 'IMPETUS' projects. Accordingly, the selection aimed to ensure good coherence with these studies also.

Thirdly, as with river flow catchment selection, an additional activity focused on
ensuring water industry relevance, both at the national scale, through consultation with
stakeholders at the eFLaG workshop, and through consultation with key demonstrator
partners (Dwr Cymru/Welsh Water and Thames Water) who identified strategically
important boreholes that would strengthen the outputs for long-term drought risk
assessment to support the water resources planning case study. Through this activity,
several additional boreholes were identified.
These selection criteria identified over 70 'candidate' boreholes for the eFLaG project.
A final quality assurance procedure was then undertaken whereby a preliminary
analysis of AquiMod's ability to capture low groundwater levels was undertaken at each
borehole via visual inspection of the simulated hydrographs. A final set of 54 boreholes
was selected (Fig. 3b). They represent a significant advance in aquifer coverage
compared to the 24 NGLA boreholes used in FFGWL, 15 of which are used in both.
**Groundwater Recharge**
The gridded groundwater recharge simulations have been aggregated over 558
'groundwater bodies' covering England (Environment Agency, 2021a), Wales (Natural
Resources Wales, 2021) and Scotland (Ó Dochartaigh et al., 2015) (Fig. 3c). These
units were used for two principal reasons. Firstly, they are physically justifiable as they
reflect known hydrogeological characteristics including groundwater recharge and
groundwater flow regimes so that each catchment represents a distinct body of
groundwater that can reasonably be considered in isolation. Secondly, they are
coherent with the licensing areas defined as part of Catchment Abstraction
Management Strategy (Environment Agency 2021b) and management areas for the
implementation of the Water Framework Directive. They are, therefore, directly
relevant to water regulation and the wider water industry.

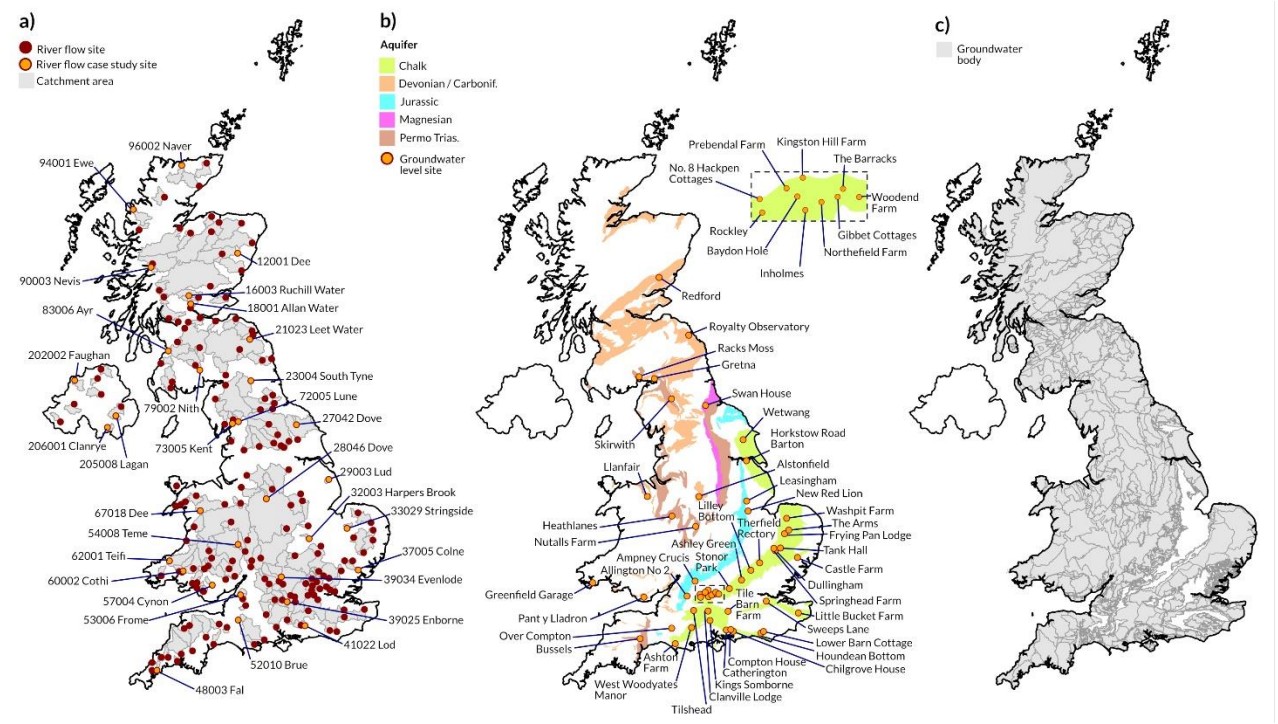



**Figure 3 a) Map of the 200 eFLaG catchments - highlighting those used as Case Study**
**sites; b) Map of 54 eFLaG boreholes and principal UK Aquifers including The Chalk,**
**Devonian and Carboniferous aquifers (Devonian/Carbonif.), Jurassic limestones**
**(Jurassic), Magnesian limestones (Magnesian) and Permo-Triassic sandstones (Permo**
**Trias.); c) Map of 558 groundwater bodies. Inset of Figure 3b shows the Berkshire**
**downs where there are a high number of boreholes.**

410

### 5. Hydrological and groundwater model ensemble setup

412

Creation of an enhanced Future Flow and Groundwater (eFLaG) dataset is
underpinned by hydrological and groundwater models used to transform rainfall and
potential evaporation (PE) to river flow, soil moisture, groundwater levels and recharge.
The approach builds on that employed under FFGWL (Prudhomme et al. 2013) whilst
exploiting developments in hydrological modelling for droughts since that time.

For modelling of river flows, eFLaG used two lumped catchment models, PDM (Moore
2007) and the GR suite (Perrin et al. 2003), and one distributed grid-based hydrological
model, Grid-to-Grid (G2G; Bell et al. 2009). PDM was used in FFWGL and therefore
provides some comparability with that project. Embracing a range of different model
structures and spatial representations can provide insights into how assessments of
future river flows (and hence, drought or low flow risk under climate change) is sensitive
to hydrological model choice. It should be noted that an important difference between
the river flow models is in treatment of artificial influences (abstractions and

discharges). G2G is not calibrated and simulates natural river flows only (i.e. it does
not include artificial influences). The GR suite and PDM do not explicitly include
artificial influences either, but as calibrated models they will implicitly include the net
effect of artificial influences in the simulations. We return to this important distinction in
the results and discussion.
For groundwater, eFLaG adopted the lumped, conceptual, AquiMod groundwater
model (Mackay et al. 2014a) to simulate groundwater level time series on a daily time
step at the boreholes identified in Section 4. AquiMod was the groundwater level model
used in FFGWL providing direct comparison. In addition to groundwater levels, the
zooming object oriented distributed recharge model (ZOODRM) (Mansour and
Hughes, 2004) was used to study changes in future groundwater recharge.
In the following sub-sections, we describe each of these models in turn, providing
information on the model set-up, calibration and past approaches to evaluation. A
consistent approach was applied to the model application and evaluation across all
these models where possible. However, it is important to emphasise that while some
aspects were common, insofar as possible (e.g. model driving data), it was necessary
to apply different approaches to suit the model in question. Calibration was done
according to past applications and best-practice. Hence, the calibration approach
described below is similar for the GR suite and PDM, but different for Aquimod, and by
its nature G2G requires no specific calibration here. Where calibration was carried out
for the conceptual models, it was undertaken for the full period of record of available
data.
Identical approaches to evaluation were adopted across all river flow models, but minor
differences applied with groundwater, as described below.
There are two sets of model output in eFLaG, described below – this terminology is
adopted throughout.
• simobs: observation-driven simulation (i.e. simulations for the observed period,
driven by observational climate datasets, described below). The simobs period
varies between models, but covers at least the January 1961 – December 2018
period.
• simrcm: UKCP18 RCM-driven simulation (12 ensemble members) (i.e.
simulations driven by the UKCP18 RCM bias-corrected dataset as described in
Section 3). These are available for 1980 to 2080. The simrcm runs from the
observed period could then be evaluated against the simobs data.
Common driving data was applied across all models for the simobs runs. Accepted
national-standard observational climate products were used, including:

- Precipitation and temperature: HadUK-Grid 1km x 1km dataset (Hollis et al. 2019), the national standard gridded meteorological dataset and observational product associated with UKCP18.
- Potential Evaporation (PE). MORECS (Hough et al., 1997), an established, national gridded PE product. Other PE datasets such as CHESS (Robinson et al., 2017) and more recently the Environment Agency's PE product (Environment Agency, 2021c) are available, however the decision to use MORECS was based on availability of data for the whole of the UK.

For all models, evaluation was undertaken in two stages, which is typical practice for appraising a model for simulation of climate change impacts:

1. Evaluation when driven with baseline observed climate data
2. Evaluation when driven with baseline climate model data.

Stage 1 involves the use of evaluation statistics to assess the performance of model simulations driven by observed climate data (the simobs runs) against observations of river flow and groundwater. For Stage 1, a range of metrics are available and widely used to assess how well rainfall-runoff or groundwater models perform against observations. Within eFLaG, these metrics were used to assess performance (Table 3). For river flows, these metrics have a focus on low flow metrics (e.g. NSE on log-transformed flows), but some do evaluate performance across the flow regime. For groundwater levels, a generalised NSE score was used which provides an overall assessment of process realism and fit to groundwater level data. The simulated and observed Standardized Groundwater level Index (SGI) were also compared using the NSE ($NSE_{SGI}$) which focusses in on groundwater extremes including droughts.

It is not possible to do a thorough evaluation of the recharge simulations from ZOODRM, given the difficulty in measuring recharge, particularly at a scale that is commensurable with a national model. However, past applications of ZOODRM (e.g. Mansour et al., 2018) have successfully used monthly river flow data as a means to evaluate ZOODRM's ability to capture catchment water balances and infer the accuracy of seasonal recharge simulations (further details provided in model description). Accordingly, a subset of the river flow metrics relevant to monthly river flows have been used to evaluate ZOODRM for stage 1.

**Table 3.** Model calibration and evaluation metrics used in eFLaG.

| Evaluation Metric | Equation | Focus |
|---|---|---|
| **Nash-Sutcliffe Efficiency (R² Efficiency)** | $$NSE = 1 - \frac{\sum_{i=1}^{n}(Q_i - q_i)^2}{\sum_{i=1}^{n}(Q_i - \overline{Q})^2}$$ $Q_i$ and $q_i$ are observed and modelled flow for day $i$ of a $n$ day record. $\overline{Q}$ is the mean observed flow. $$NSE = 1 - \frac{\sum_{i=1}^{n}(H_i - h_i)^2}{\sum_{i=1}^{n}(H_i - \overline{H})^2}$$ $H_i$ and $h_i$ are observed and modelled groundwater level for day $i$ of a $n$ day record. $\overline{H}$ is the mean observed groundwater level. | High Flows/Generalised groundwater levels |
| **Nash-Sutcliffe Efficiency log flows*** | $$NSE_{log} = 1 - \frac{\sum_{i=1}^{n}(\log(Q_i) - \log(q_i))^2}{\sum_{i=1}^{n}(\log(Q_i) - \overline{\log(Q)})^2}$$ | Low Flows |
| **Nash-Sutcliffe Efficiency square root flows** | $$NSE_{sqrt} = 1 - \frac{\sum_{i=1}^{n}(\sqrt{Q_i} - \sqrt{q_i})^2}{\sum_{i=1}^{n}(\sqrt{Q_i} - \overline{\sqrt{Q}})^2}$$ | Generalised Flows |
| **Nash-Sutcliffe Efficiency standardised groundwater level index** | $$NSE_{SGI} = 1 - \frac{\sum_{i=1}^{n}(SGI_i - sgi_i)^2}{\sum_{i=1}^{n}(SGI_i - \overline{SGI})^2}$$ $SGI_i$ and $sgi_i$ are observed and modelled SGI for day $i$ of a $n$ day record. $\overline{SGI}$ is the mean observed SGI. | Groundwater extremes |
| **Modified Kling Gupta Efficiency [square root flows]** | $$KGE'_{sqrt} = 1 - \sqrt{(r-1)^2 + (\beta-1)^2 + (\gamma-1)^2}$$ where $r$ is the correlation coefficient, $\beta$ is the bias ratio $\frac{\mu_{\sqrt{q}}}{\mu_{\sqrt{Q}}}$, and $\gamma$ is the variability ratio $\frac{CV_{\sqrt{q}}}{CV_{\sqrt{Q}}}$ or $\frac{\sigma_{\sqrt{q}}/\mu_{\sqrt{q}}}{\sigma_{\sqrt{Q}}/\mu_{\sqrt{Q}}}$ $\mu$, $\sigma$ and $CV$ are the mean, standard deviation and coefficient of variation of flow (here of the square root of modelled and observed flows as indicated by the suffix) | Generalised flows |

| | | |
|---|---|---|
| **Absolute Percent Bias** | $absPBIAS = \left\| \dfrac{\sum(q_i - Q_i)}{\sum Q_i} \right\| 100$ | Water Balance |
| **Mean Absolute Percent Error** | $\text{MAPE} = \left( \dfrac{1}{n} \sum\limits_{i=1}^{n} \left\| \dfrac{Q_i - q_i}{Q_i} \right\| \right) 100$ | Systematic |
| **Absolute Percent Error in Q95** | $Q95_{APE} = \left\| \dfrac{Q95 - q95}{Q95} \right\| 100$ | Low Flows |
| **Low Flow Volume** | $LFV = 100 \dfrac{\sum_{p=70}^{95}(\sqrt{q_p} - \sqrt{Q_p})}{\sum_{p=70}^{95}(\sqrt{Q_p})}$<br><br>Here $q_p$ and $Q_p$ are the modelled and observed flow $p$ percentiles | Low Flows |
| **Absolute Percent Error in the Mean Annual Minimum on a 30-day moving average\*** | $MAM30_{APE} = \left\| \dfrac{QMAM30 - qMAM30}{QMAM30} \right\| 100$<br><br>where $QMAM30$<br>$= \dfrac{1}{n} \sum\limits_{j=1}^{n} \min_j \left( \dfrac{Q_{j,i-29} + Q_{j,i-28} + Q_{j,i-27} \dots Q_{j,i-1} + Qj_{,i}}{30} \right)$<br><br>Here $Q_{j,i}$ is observed flow for day $i$ of hydrological year $j$ for a record of $n$ years | Low Flows |

\*1/100th of the mean observed flow was added to both modelled and observed flow values during evaluation in order to avoid errors and biases due to very small and zero flows.

501

502

503

Sources of quality controlled, long-term observational data for model calibration and evaluation were the national standard repositories for hydrological data:

- River Flows: UK National River Flow Archive https://nrfa.ceh.ac.uk/
- Groundwater Levels: UK National Groundwater Level Archive https://www2.bgs.ac.uk/groundwater/datainfo/levels/ngla.html

Stage 2 appraises the performance of the models when driven by the climate model outputs. That is, it compares the simobs and simrcm runs over the common baseline period. This assessment cannot use performance metrics based on time-series, as climate models are not expected to reproduce the sequencing of events seen over the historical period (Kay et al.

2015). Instead, the comparison has been done in terms of river flow and groundwater level
duration curves, low flow/level metrics and seasonal recharge values. Thus, comparing the
statistical characteristics of river flows, groundwater levels and groundwater recharge rather
than their day-to-day equivalence (Kay et al. 2015, 2018). When looking at the performance
of an ensemble of climate model runs, the model simulation driven by observed data would
ideally sit within the range covered by the ensemble (assuming an ensemble of sufficient
size). However, it would not necessarily be expected to sit in the middle of the ensemble
range, because the set of weather events that actually occurred within the historical observed
baseline period is just one realisation of what could have occurred within the range of natural
variability (Kay et al. 2018).

## Description of the models and specific setup

### GR4J/GR6J

The GR4J and GR6J models come from a suite of hydrological models provided in the
"airGR" modelling suite (Coron et al. 2021) for the R software programme. Both models are
well suited to application across many catchments using the inbuilt automatic parameter
optimisation function. The simple, efficient form of airGR models also make them suitable for
uncertainty and ensemble analyses.
GR4J (Génie Rural à 4 paramètres Journalier) is a simple daily lumped conceptual model
with only four free parameters. GR4J has been used for hydro-climate change research
across the globe, and has demonstrated good performance in a diverse set of catchments in
the UK. The model has been applied in the UK for operational seasonal forecasting, as well
as for long-term drought reconstructions nationwide (Harrigan et al. 2018b, Smith et al.
537 2019).

GR6J (Génie Rural à 6 paramètres Journalier) (Pushpalatha et al. 2011) is a six parameter
variant of the GR modelling suite that was developed to improve low flow simulation and
groundwater exchange. Recently, GR6J has increasingly been applied in UK water resources
applications (e.g. Anglian Water Drought Plan, 2021).
For eFLaG, it was decided, therefore, that using both GR4J and GR6J would be beneficial.
Both GR4J and GR6J were calibrated using the inbuilt automatic calibration function, with the
modified Kling Gupta Efficiency (KGE, Gupta et al, 2009; Kling et al 2012) as the Error
criterion ('ErrorCritKGE2'). KGE offers a thorough error criterion as it calculates the
correlation coefficient, the bias and the variability between simulated and observed flows.
KGE values range from –Inf to 1, with 1 being a perfect fit. The calibration algorithm was
applied to square-root transformed flows in order to place weight evenly across the flow
regime. The airGR snowmelt module "CemaNeige" was not applied, as a simple snow
module was applied to the climate data to pre-process the precipitation data into rainfall and
snowmelt based upon temperature (See section 3).

**Grid-to-Grid**

The Grid-to-Grid (G2G) hydrological model is an established area-wide distributed model that
has been used to investigate the spatial coherence and variability of floods and droughts at
catchment, regional and national scales. Model output typically consists of natural river flows
at both gauged and ungauged locations, and can be provided as time-series for specific
locations as well as 1km x 1km grids. The G2G has been used for climate impacts modelling
of floods (Bell et al., 2009, 2012), low flows (Kay et al., 2018) and droughts (Rudd et al., 2019)
and is also used operationally for flood forecasting (Cole and Moore, 2009; Moore et al.,
560 2006).

The G2G is typically configured on a 1km×1km grid across Great Britain using spatial
datasets of landscape properties such as soil type and drainage network, together with a few
nationally-applied model parameters. The model is thus parameterised using national-scale
spatial datasets (e.g. soil grids), rather than via individual catchment calibration. The spatial
datasets and parameters used here are the same as those used in previous studies (Rudd
et al., 2019; Bell et al., 2009, 2012; Kay et al., 2018). Note that model output for G2G is for
186 of the 200 eFLaG catchments. Of the 14 catchments excluded, 9 are in Northern Ireland
and so not covered by the version of G2G applied here. For the other five catchments there
were difficulties identifying appropriate outlet locations on the 1km network of flow directions
used by G2G.
The G2G can either be initialised with model water stores set to default or zero values, or
from a states file appropriate to the run start date. In eFLaG the G2G was run for two years
with observed rainfall and PE to provide a 1 January 1963 states file to initialise the
observation-driven G2G model run. The RCM-driven G2G runs were all initialised with a
generic December states file provided by an obs-driven run (for 1 December 1980), then the
first two years of each RCM-driven run were discarded to allow for model spin up. The eFLaG
river flow datasets therefore cover the periods, 1 January 1963 to 31 December 2018
(simobs) and 1 December 1982 to 30 November 2080 (simrcm).

**PDM**

The Probability Distributed Model or PDM (Moore, 2007; UKCEH, 2021) is a simple, very
widely used lumped rainfall-runoff model that can be configured to a variety of catchment flow
regimes. Within the model, a soil water store with a distribution of water absorption capacities
controls runoff production through a saturation excess process; stored water is also lost to
evaporation. In one configuration, all runoff enters a surface store (the fast pathway) while a
groundwater store (the slow pathway) is recharged by soil water drainage. In an alternative
configuration, the runoff is split between the two stores according to a fixed fraction. Water in
the surface- and ground-water stores is routed using a non-linear storage equation (powers
of 1, 2 and 3 were trialled under eFLaG), or, for the surface store, a cascade of two linear
reservoirs, before being combined to produce the modelled flow at the catchment outlet.
Water is conserved within the model, whilst a multiplicative factor (equal to 1 if not required)
is applied to the input precipitation. Alternatively, a Groundwater Extension (Moore and Bell,
2002) may be invoked to allow modelling of underflow at the catchment outlet, external
springs, pumped abstractions, and the incorporation of well level data. Multiple hydrological
response zones within a catchment can also be represented (not trialled under eFLaG). PDM
may be thought of as a toolkit of model components representing a range of runoff production
and flow routing behaviours, and with a choice of time-step.
Under eFLaG, single zone PDM models were invoked with a daily time-step. The model
stores were initialised using the mean observed flow over the period of record, and the first
two years of model flow discarded to allow for model spin-up. Nineteen different combinations
of the above-mentioned toolkit options were systematically trialled for each catchment.
Parameter estimation was performed using an automatic calibration procedure that applied
a simplex optimisation scheme (Nelder and Mead, 1965) to increasing numbers of model
parameters in turn. The rainfall factor, or, when employed, a spring factor (representing net
water exchange for the catchment), were used to achieve zero bias in the modelled flows
with respect to observations. Remaining parameters were estimated so as to optimise the
modified Kling-Gupta Efficiency calculated on either the square root transformed flows, or, to
a limited extent, the log transformed flows (Supplementary info S.2).

**AquiMod**

AquiMod is a lumped conceptual groundwater model that links simplified equations of soil
drainage, unsaturated zone flow, and saturated groundwater flow to simulate daily
groundwater level time series at a specified borehole (Mackay et al., 2014b). Each of these
three components use model parameters that describe site-specific hydrological and
hydrogeological characteristics of the groundwater catchment surrounding the borehole. The
model also has a flexible saturated zone model structure that can be modified to represent
different levels of vertical heterogeneity in hydrogeological properties.
For each borehole, the AquiMod parameters and structure were calibrated to achieve the
most efficient simulation of available historical groundwater level data using the Nash-
Sutcliffe Efficiency (NSE), which provides a reliable assessment of overall process realism
and goodness of fit to groundwater level time series; following the approach of Mackay et al.
(2014a) and Jackson et al. (2016), model parameters that could be related to catchment
information (e.g. relating to known land cover and soil type) were fixed. The remaining
parameters were then calibrated, using six different saturated zone model structures
including a one-layer model (fixed hydraulic conductivity and specific yield); two- and three-
layer models with variable hydraulic conductivity and fixed specific yield; two- and three-layer
models with variable hydraulic conductivity and variable specific yield; and a 'cocktail glass'
representation of hydraulic conductivity variation with depth (Williams et al., 2006). The
optimal structure-parameter combination was obtained for each borehole using the Shuffled
Complex Evolution global optimisation algorithm.
The calibrated models were then evaluated for their ability to capture groundwater level
extremes using the Standardized Groundwater level Index, SGI (Bloomfield and Marchant,
2013) as the basis for this evaluation. The SGI is a normalised index, calculated directly from
groundwater level time series, which can be used to identify droughts and provide a
quantitative status of groundwater resources drought events (e.g. Bloomfield et al., 2019).

## ZOODRM

ZOODRM is a distributed recharge calculation model originally developed to estimate
recharge values to drive groundwater models (Mansour and Hughes, 2004). It is applied over
the British Mainland using a 2km square grid. The FAO Drainage and Irrigation Paper 56
(FAO, 1988) approach, modified by Griffiths et al. (2006), is used to calculate potential
recharge. This method removes actual evaporation and soil moisture deficit from rainfall and
calculates potential recharge as a fraction of the excess water using a runoff coefficient value.
The model was driven by daily rainfall and potential evaporation data. The model was
primarily parameterised using available national scale data including data relating to the soil
hydrology (Boorman et al., 1995), vegetation (LCM2000, NERC) and surface topography.
The latter of these was used to route surface water runoff.
The runoff coefficient, which defines the proportion of excess soil water that drains overland
via surface runoff, is an unknown parameter which must be calibrated. This was done in two
stages. Firstly, the calibration problem was simplified by defining zones of equal runoff
coefficient. In total 35 zones were used in ZOODRM which were based on UK
hydrogeological and geological maps (DiGMapGB-625, 2008). Then, the runoff coefficient
for each zone was manually calibrated by comparing simulated runoff to observed river flows
minus baseflow which was calculated using a well-established baseflow separation method
(Gustard et al., 1992). This was done using monthly mean flows given that ZOODRM does
not have a sophisticated runoff routing scheme, and it is not expected, therefore, to capture
daily variability in runoff. The comparison to monthly flows does, however, provide a useful
means to evaluate the seasonal water balance of the model which serves as the best
available proxy for the accuracy of the recharge simulations. In total, 41 gauging stations
were used to assess the model performance.
The only hydrological process that needs initialisation in the ZOODRM is the soil moisture
deficit. As all simulations start in January, which is a wet month with minimal potential
evaporation, it is assumed that the initial soil moisture deficit is equal to zero. Even so, a
warm up period of one year is used to initialise the model.

**6. Hydrological model evaluation (Stage 1 evaluation)**

This section provides a brief summary of the outputs of the Stage 1 evaluation. Note that for
river flows, model evaluation was undertaken at the same gauged locations and for the same
period of time used for model calibration, except G2G which is not specifically calibrated.
**River Flows**
Fig. 4 summarises the range of Stage 1 evaluation metrics across all catchments, while
Supplementary Figs S2 to S5 provide maps of the evaluation metrics at each catchment. For
GR4J, generally there was good performance across performance metrics in most
catchments. Some outliers are present in the drought metrics, particularly in the South East
and London. For GR6J, we observed good performance across all performance and drought
metrics. GR6J generally performs slightly better than GR4J, particularly as shown in low flow
catchments in the logNSE metric. For PDM, very good scores are obtained across the 200
sites, especially the low flow/drought indicators (bottom rows).
For G2G, again, good performance was observed overall (medians for NSE/ logNSE/
sqrtNSE/ KGE2 ≥ 0.7). However, the performance was generally lower than for GR or PDM
because the G2G is not calibrated to individual catchments, and G2G simulates *natural* flows,
whereas the lumped models are calibrated to the observations used for performance
assessment. In catchments with a high degree of anthropogenic disturbance, G2G is less
able to simulate observed flows, whereas the calibration of the other hydrological models will
implicitly account for such artificial impacts, meaning they are inevitably more likely to
replicate observed flows, even if these processes are not included explicitly.
This distinction highlights an important benefit of eFLaG: PDM and GR4J/GR6J are calibrated
to present-day flows and hence simulated flows are not natural, as they implicitly include
artificial impacts. These runs do not, therefore, allow users to separate natural flows and
artificial influences in the baseline period, nor to project how they may change relative to each
other in future. On the other hand, although not used here, G2G has the capability of including
artificial influences separately (e.g. Rameshwaran et al., 2022). We return to this issue in
Section 8.






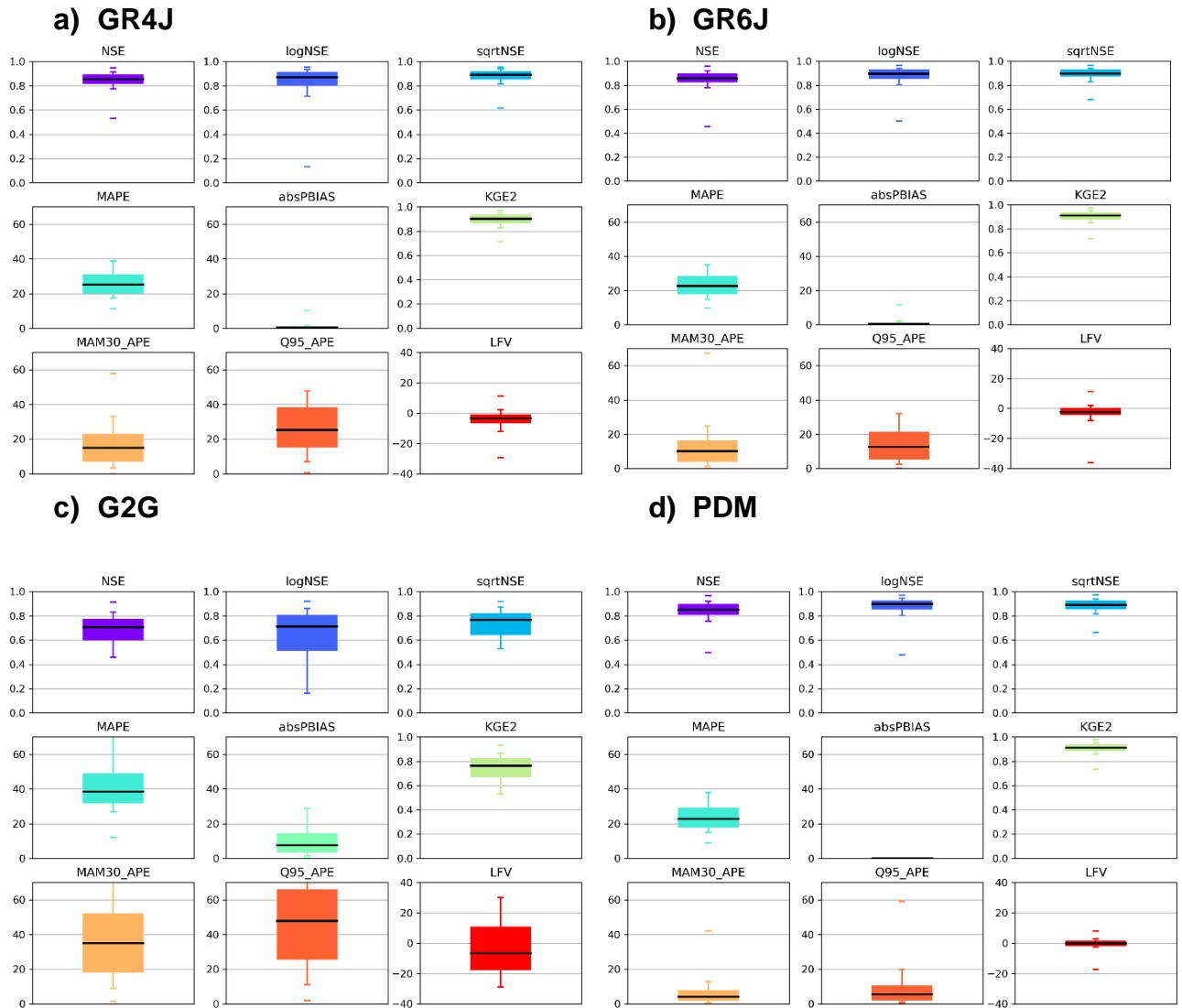

**Figure 4:** Evaluation of modelled river flow performance. The key evaluation metrics outlined in Table 3 are summarised for all 200 modelled catchments (GR4J, GR6J, PDM), or 186 modelled catchments (G2G).

In general, the eFLaG dataset shows a very good range of performance comparable with previous applications of these models for the UK (e.g. Rudd et al. 2017; Harrigan et al. 2018b; Smith et al. 2019). There are some commonalities with these previous studies in terms of spatial patterns. Rudd et al. (2017) also noted that G2G performance is likely to reflect the fact that simulated flows are natural (hence performance is poorer in the south and east where artificial influences are typical greater). Issues with poorer performance in groundwater-dominated catchments were highlighted for GR4J by Smith et al. (2019) and the

present study shows that eFLaG enables some improvement through GR6J. Smith et al.
(2019) also highlighted how a lack of snowmelt constrained performance in some areas (e.g.
NE Scotland) while the current results also show improvements in these areas in eFLaG,
given the inclusion of snowmelt accounting.

**Groundwater levels**
Fig. 5 summarises the model evaluation results for the 54 AquiMod models used in eFLaG.
The results show that all 54 models demonstrate good overall efficiency in capturing daily
groundwater level dynamics, achieving a NSE ≥ 0.77. All but 11 of the models achieve a NSE
≥ 0.85 and 28 of the models achieve a NSE ≥ 0.90. These include all 7 models situated in
the Permo-Triassic sandstone and 4 out of 5 of the models situated in the Devonian and
Carboniferous aquifers. Swan house and Lower Barn Cottage; the only models situated in
the Magnesian limestones and Lower Greensand respectively, achieved a NSE of 0.82 and
0.86. The Chalk and Jurassic limestones borehole models span the full range of NSE scores.
The results show that all 54 AquiMod models are able to capture the historical SGI time series
efficiently, achieving a $NSE_{SGI}$ ≥ 0.6 which indicates that the models effectively capture
groundwater extremes including periods of drought. The majority of models show a lower
$NSE_{SGI}$ compared to the NSE, although several models show negligible difference. On
average the $NSE_{SGI}$ is 0.15 less than the NSE.

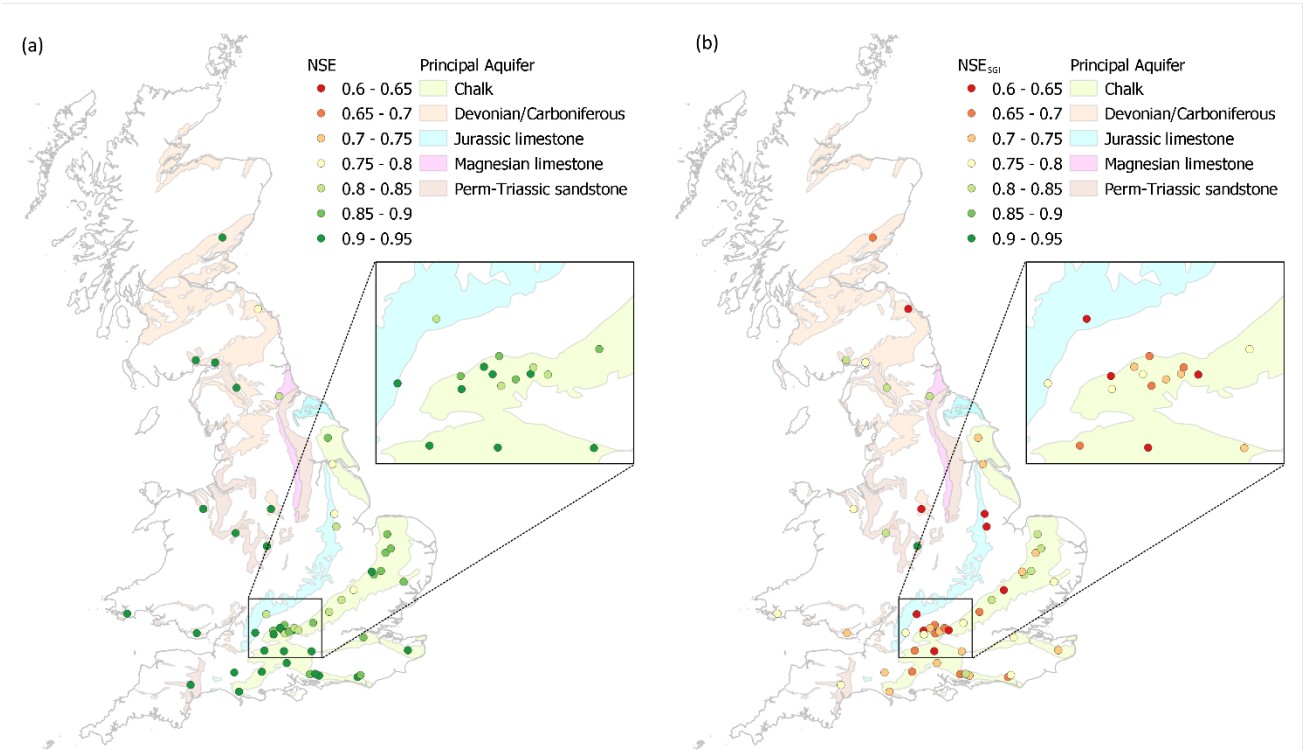

**Figure 5: AquiMod evaluation metric results including NSE (a) and $NSE_{SGI}$ (b).**

**Groundwater recharge**

ZOODRM demonstrates an ability to efficiently capture monthly mean river flows as is reflected by the medians for NSE and KGE2 which both exceed 0.75 and the median absolute percent bias which is 12.7% (Fig. 6). Fig. S6 shows the distributed recharge model results at the 41 gauging stations across the country. The model uses a simplistic overland routing approach, which is implemented to check the water balance at a monthly basis, noting that large scale spatial recharge values are most commonly used to drive groundwater flow models using monthly stress periods.

**NSE**                    **MKGE**                    **absPBias**

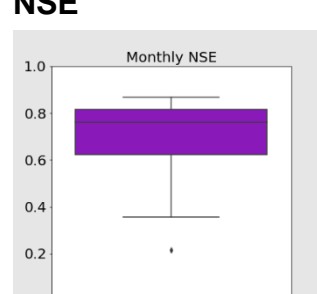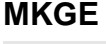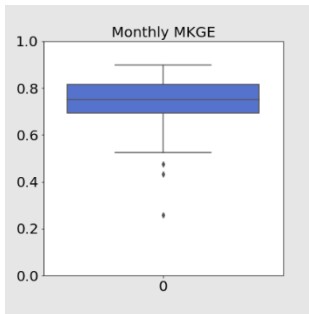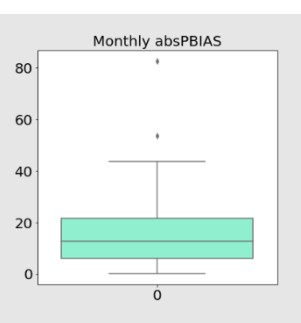

**Figure 6: Distributed recharge model ZOODRM evaluation results.**

### 7. Evaluation of RCM-based runs in the baseline

This section briefly considers the outcomes of the Stage 2 evaluation, focusing firstly on flow/groundwater duration curves for a subset of eFLaG sites, and then specifically on representation of particular low flows (low groundwater level) quantiles.

**Flow duration curves**

Flow duration curves (FDCs) summarise the entirety of the flow regime from high to low flows by including all river flows and expressing them in terms of the percentage of time a given flow is exceeded. Fig.7 and Figs. S7 to S9 provides a perspective on the ability of the RCM-driven river flow simulations (simrcm) to replicate the range and frequency of flows based on the observation climate-driven river flow simulations (simobs). FDCs are shown for a common baseline period of 1989-2018

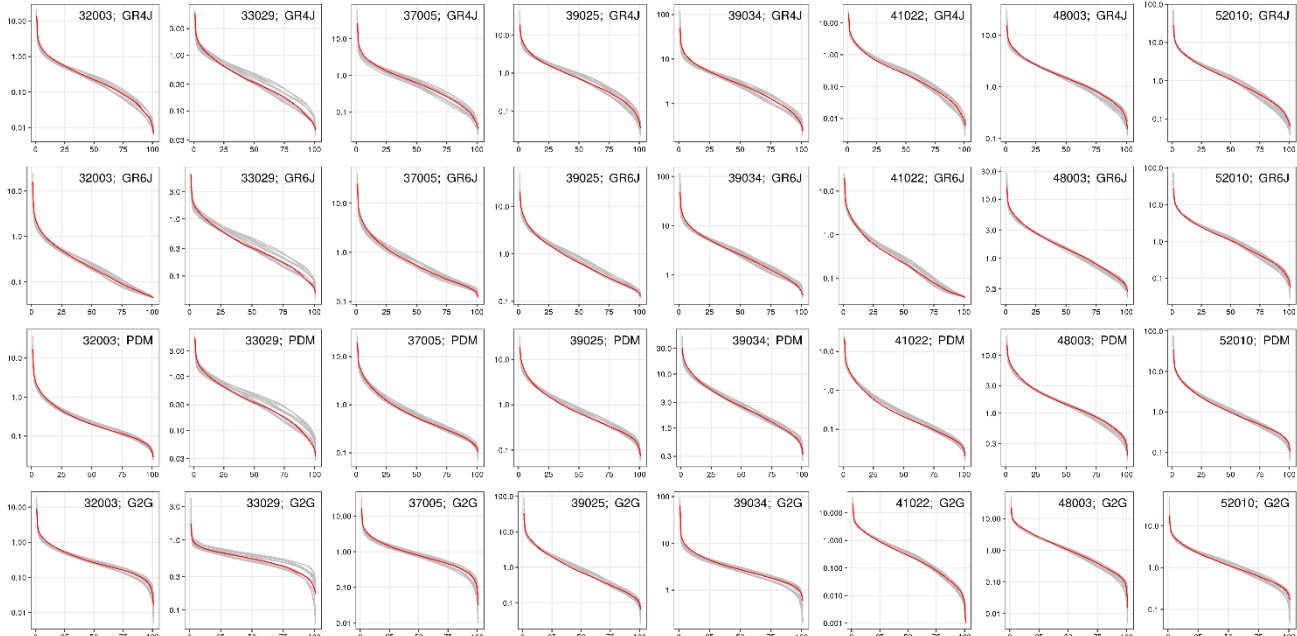

**Figure 7 -- Flow duration curves (FDCs) comparing the baseline flow regime in the 12 RCM ensemble members (simrcm, grey lines) to simulated observed (simobs, red line), 1989-2018. FDCs are featured for four hydrological models (GR4J, GR6J, PDM, G2G; rows) and eight catchments in southern and eastern England (32003 Harpers Brook, 33029 Stringside, 37005 Colne, 39025 Enborne, 39034 Evenlode, 41022 Lod, 48003 Fal, 52010 Brue; columns). The y-axis represents river flows (cumecs) on a logarithmic scale.**

The close correspondence between FDCs derived from the RCM ensemble members and model observations suggests that the RCM ensemble is performing well in replicating flows across the regime This is consistent across most UK catchments, illustrated by the representative subset of 32 catchments featured in Fig. 7 and Figs.S7 to S9. The model observations are usually within the range of values from the 12 ensemble members throughout the flow regime.  There are some catchments for which the RCM ensemble is more likely to overestimate the lowest half of the flow regime (exceedance probabilities of 50-100), most notably for the Stringside (33029; Fig.7), Dove (28046; Fig.S7), Frome (53006; Fig. S8), and Lud (29003; Fig. S7).

For certain catchments such as the Stringside (33029; Fig. 7) and Lud (29003; Fig. S7), although there appears to be greater RCM uncertainty in river flows than for other catchments, the differences tend to be exaggerated in smaller, drier catchments with lower flows across the flow regime.  The logarithmic *y*-axis is also a contributing factor to this, and also accounts for the seemingly larger RCM uncertainty in low flows than high flows across all catchments. These findings are also consistent across the four hydrological models, with no systematic differences identified for a given hydrological model. In some exceptional circumstances, there are examples of certain models in specific catchments in which the lowest river flows derived from the RCM ensemble are much lower than those in the model

observations (e.g. 23004 South Tyne (Fig. S7) and 67018 Welsh Dee (Fig. S8) for GR6J,
33029 Stringside (Fig. 7) for G2G).

**Groundwater level duration curves**

Overall, an analysis of the groundwater level duration curves (GLDCs) at all boreholes
(Figs.S10-S15) shows close correspondence between the simrcm and simobs runs whereby
the simobs GLDC typically lies within the range of the simrcm GLDCs. However, there are
some different behaviours across the boreholes which are summarised in Fig. 8. Fig.8a
shows the GLDCs for the New Red Lion borehole situated in the Lincolnshire Limestone, the
results of which are representative of most boreholes where the majority of simobs GLDCs
falls within the range of the simrcm GLDCs. Several of the boreholes show a relatively high
degree a variability across the simrcm runs in comparison to the simobs including the
Heathlanes borehole situated in the Permo-Triassic Sandstone (Fig. 8b). These appear to be
associated with boreholes which are known to respond relatively slowly to climate due to local
hydrogeological conditions. For example, Heathlanes is known to be representative of a
relatively low hydraulic diffusivity aquifer. For some boreholes there are areas of the GLDCs
where the simobs GLDC does not lie within the range of the simrcm GLDC. In the most
extreme cases, systematic biases across almost the entire GLDC can be seen (e.g. Fig. 8c).

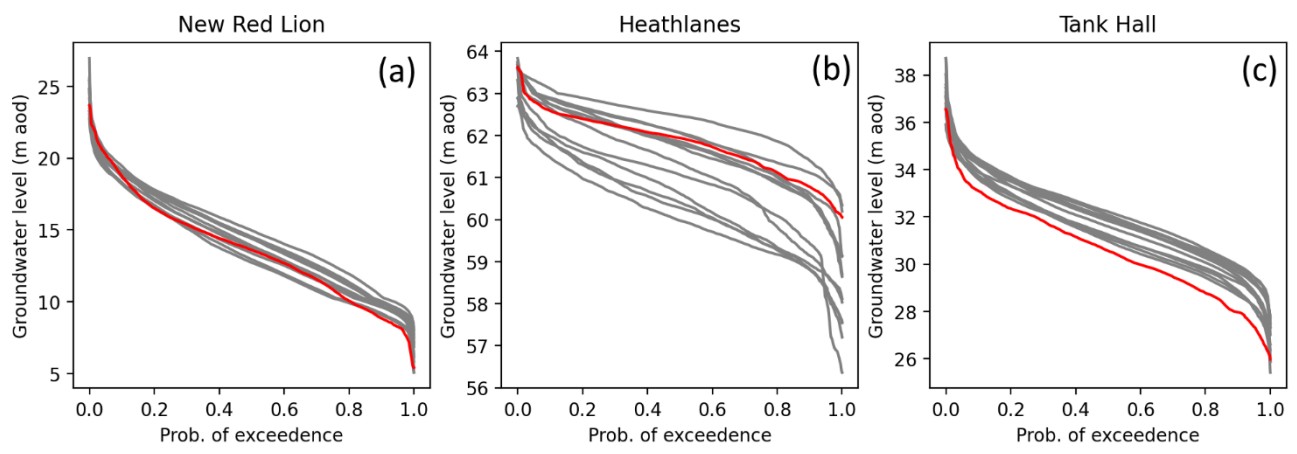


**Figure 8 – Groundwater level duration curves (GLDCs) for the period 1989-2018 using the**
**simrcm (grey lines) simobs (red line) simulations. GLDCs are featured for three boreholes in**
**different hydrogeological settings which show contrasting behaviour: (a) New Red Lion,**
**(Lincolnshire Limestone), (b) Heathlanes (Permo-Triassic sandstone, Shropshire), (c) Tank**
**Hall (Chalk).**

**Low river flows and groundwater levels**

Replication of observed low river flows and groundwater levels over a baseline period
provides an indication of how well the simrcm runs are performing at the lower part of the
river flow and groundwater level regime, and therefore enhances confidence in future low
flow and level projections. Figs 9a-d show the difference between the simobs and simrcm
90% exceedance flow (Q90) over the 1989-2018 baseline period reported as absolute
percentage error (APE) at each of the 200 catchments for all four river flow models.

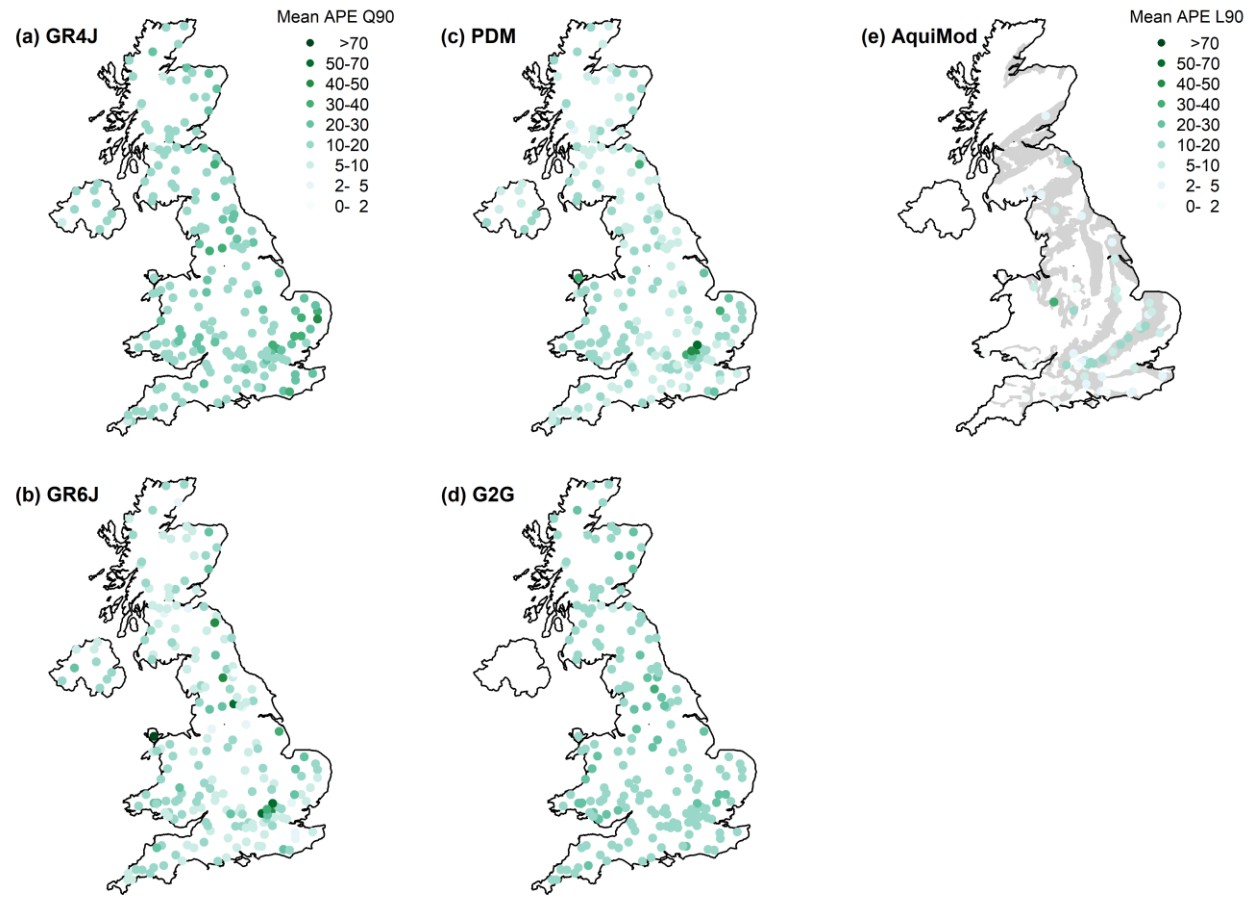


**Figure 9 -- Comparison of simobs and simrcm runs for river flows and groundwater levels
exceeded 90% of the time (Q90 and L90 respectively) between 1989 and 2018. Colour scale
indicates the mean of 12 absolute percent errors (APEs) between Q90/L90 in model
observations and Q90/L90 in each of 12 ensemble members. Results are presented for: (a)
GR4J; (b) GR6J; (c) PDM; (d) G2G; (e) AquiMod.  Note: AquiMod levels are expressed as a
percentage of the simobs range in groundwater levels to remove the influence of aquifer
storage.  Figures S16 to S18 feature the equivalent baseline assessment for Q30/L30, Q50/L50
and Q70/L70.**

Overall, there is a reasonable agreement between the simobs and simrcm Q90 values across
all four models. Mean APEs are less than 20% for most catchments across the four
hydrological models.  Modelled low flows for GR6J, G2G and particularly PDM are especially
well replicated in catchments across the UK, with mean APEs higher (20-50%) in GR4J river
flows for catchments in East Anglia and parts of northern England and south Wales.  The
lumped catchment models GR6J and PDM struggle to capture low flows in groundwater-
influenced catchments of the east Chilterns north of London, with APEs of up to 70%.
Considering the natural flows simulated by G2G and the prevalence of artificial influences on

rivers further south and east in the UK, mean APEs are reasonable in this region and are actually higher in more natural parts of Wales and northern England.

Mean APEs at a range of other flow quantiles demonstrate similar patterns (Figs S16 to S18). Mean APEs of Q30 for the vast majority of catchments for all four hydrological models are less than 20% (Fig. S16). Mean APEs of Q50 (Fig. S17) and Q70 (Fig. S18) are also reasonable in most catchments and models, though higher mean APEs (20-50%) are apparent for both of these flow quantiles in East Anglia for GR4J, in parts of northern England for G2G, and in groundwater-influenced parts of the Chilterns for PDM. Mean APEs are similarly higher in GR6J flows at Q50 in East Anglia and at Q70 in the groundwater-influenced Chilterns. Whilst this analysis is primarily an assessment of the ability of the RCM ensemble to replicate flows across the regime, it is clear that the hydrological model calibrations also have a role in influencing the outcomes.

Fig. 9e shows the difference between the simobs and simrcm 90% exceedance groundwater level (L90) over the 1989-2018 baseline period reported as absolute percentage error (APE) relative to the simobs range in groundwater levels at each of the 54 boreholes. The use of the range in groundwater level as a reference removes the influence that the aquifer storage has on groundwater variability across the boreholes. There is good agreement between the simobs and simrcm L90 values across the boreholes. Mean APEs are less than 20% for all of the boreholes except for the Heathlanes borehole in the Permo-Triassic Sandstone where Mean APE exceeds 30%.

Mean APEs at a range of other groundwater level quantiles demonstrate similar patterns (Figs S16 to S18). Mean APEs of L30 do not exceed 5% for the majority of boreholes. The mean APE's typically become larger for most boreholes as the level quantile reduces towards L90. Heathlanes consistently has the highest mean APE for all level quantiles.

**Seasonal groundwater recharge**

Fig. 10 provides a comparison of simobs and simrcm runs for seasonal average groundwater recharge between 1989 and 2018 generated by ZOODRM. During the winter months (DJF), when groundwater recharge is highest, the simrcm simulations show good correspondence with simobs simulations where the mean APE is less than 20% for all, but seven of the groundwater bodies. During the summer months (JJA), when groundwater recharge is lowest, the majority of groundwater bodies still show mean APE of less than 20%, but over 200 of them show errors exceeding 20%. These larger errors are typically associated with groundwater bodies that have lower than average recharge for this time of year. For MAM, the majority of groundwater bodies with errors that exceed 20% are also associated with those GW bodies with below-average recharge for that time of year. There are also some additional areas with significant recharge that show errors exceeding 20% including groundwater bodies in eastern-central Scotland, north-west and south-west England. For autumn (SON), the simrcm simulations show good correspondence with simobs simulation

where the majority (>80%) of groundwater bodies show a mean APE of less than 20%. The
majority those with larger errors are situated on the east coast of Scotland and England, north
Wales and Cheshire.

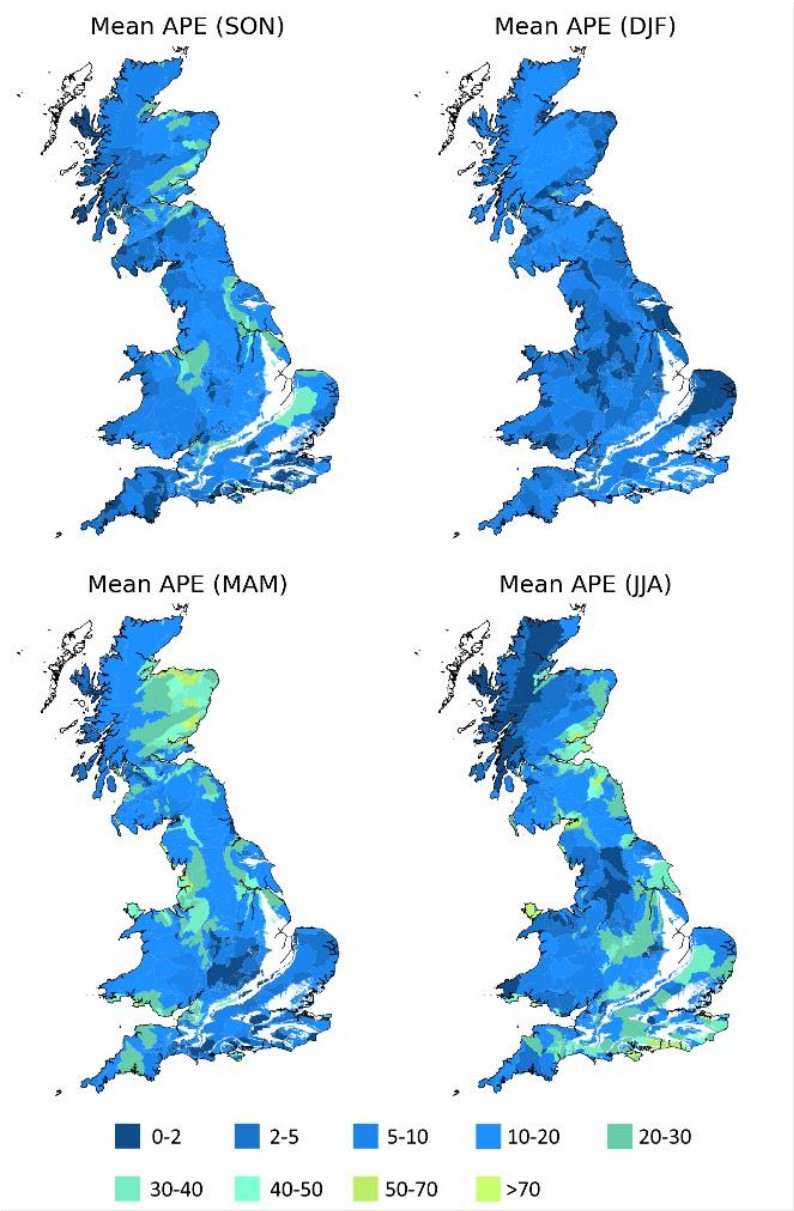


**Figure 10 -- Comparison of simobs and simrcm runs for seasonal average groundwater**
**recharge between 1989 and 2018 generated by ZOODRM. Colour scale indicates the mean of**
**12 absolute percent errors (APEs) between simobs and simrcm.**



## 8. Applications and limitations

**Applications**

The eFLaG dataset is presented as a nationally consistent dataset of future river flow, groundwater and groundwater recharge, using the latest available climate projections, from UKCP18. In this article, we have described the dataset and its evaluation against observational hydrological datasets, to give some confidence in the use of eFLaG as a dataset that can be used to assess the potential impacts on climate change on UK hydrology for a very wide range of applications.

The eFLaG dataset was developed specifically as a demonstration climate service for use by the water industry for water resources and drought planning, and hence by design is focused on future projections of drought, low river flows and low groundwater levels. We therefore present eFLaG primarily as a dataset for this purpose. Ongoing work is underway to demonstrate the utility of eFLaG for future drought projections (Parry et al. submitted; Tanguy et al. submitted) and for future drought/water resources planning in practice (Counsell et al. in prep.). The predecessor product, FFGWL, has been widely used within the water industry to provide insight into the future evolution of river flows and groundwater levels through the 21st century to support water resources management plans, and also supported significant academic water resource planning studies (e.g. Borgeomo et al. 2015; Huskova et al. 2016).

To provide users with a platform for accessing eFLaG datasets, and all the evaluation approaches outlined here, an interactive web application has been developed, the eFLaG Portal (https://eip.ceh.ac.uk/hydrology/eflag/). The Portal provides a user friendly front-end for accessing eFLaG results, with several examples shown in Fig 11. The figure demonstrates how eFLaG data can be used to project future drought characteristics for various timeslices, and also how low flow characteristics change through the 21st century, based on the analysis conducted in Parry et al. (submitted).




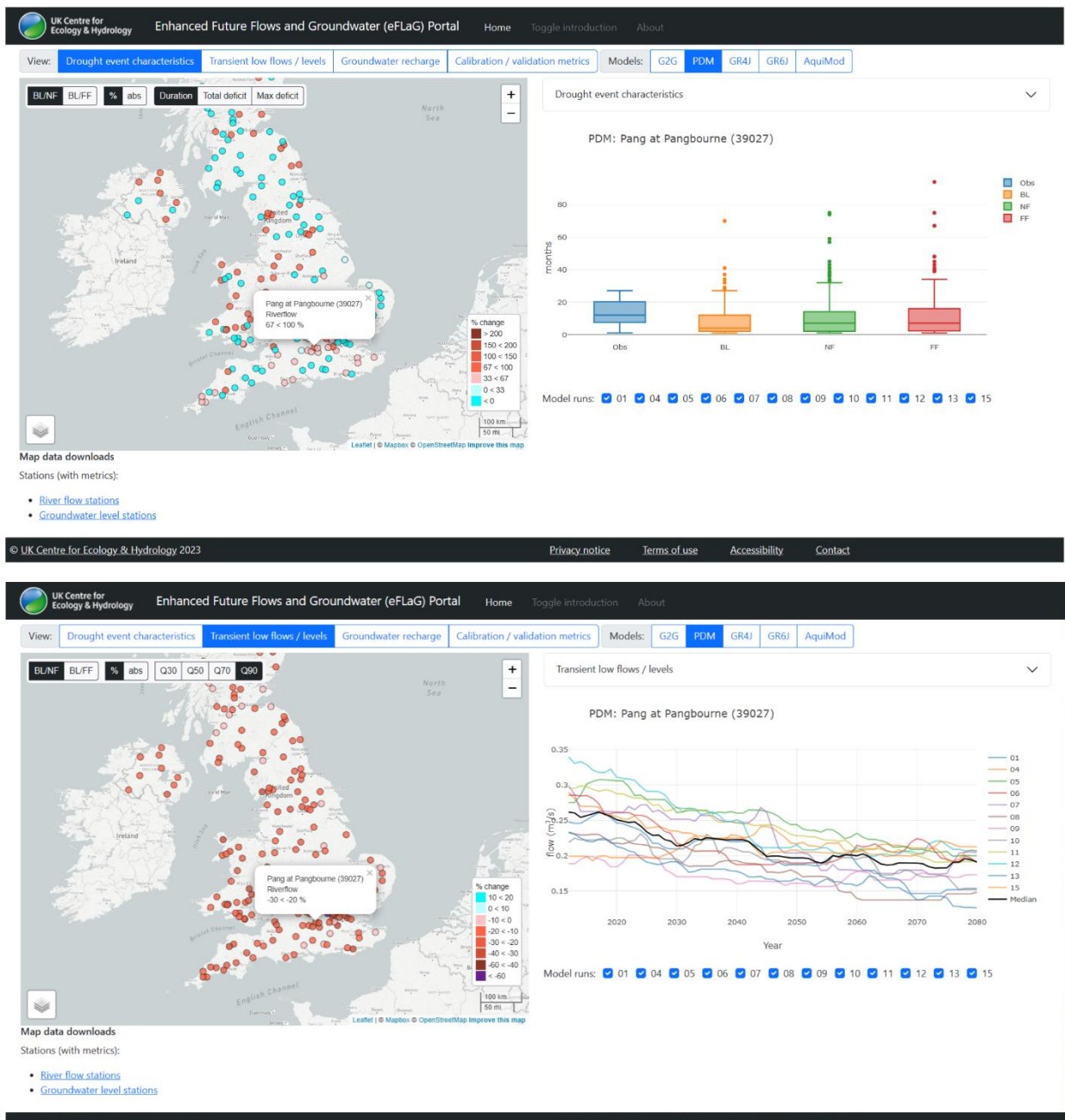


**Figure 11: screenshots from the eFLaG Portal. Top: map showing percentage change in drought duration between baseline and near future for eFLaG catchments nationally, using PDM; boxplots showing % changes (using PDM) for a river in southern England (the river Pang) for three timeslices, with boxplots showing range of RCM uncertainty; other drought characteristics available on other tabs. Bottom: map showing percentage change in a low flow metric (Q90) between baseline and near-future for eFLaG catchments nationally, using PDM;**

**with time series showing transient projections of Q90 in moving windows through to the 2080s for the river Pang, each colour representing different RCM runs, black representing median. For all outputs, models other than PDM can be selected using the tabs at the top.**

By providing a consistent dataset of future river flows, groundwater levels and groundwater recharge, eFLaG can potentially support a wide range of applications across other sectors. The FFGWL product also found very wide application for diverse research purposes (for: water quality, e.g. Charlton et al. 2018; hydroecology, e.g. Royan et al. 2016; groundwater recharge, Hughes et al., 2021; groundwater level reconstruction, Jackson et al., 2016). For eFLaG, the good simulation of river flows and groundwater behaviours across much of the hydrological range suggests that this product could also find application in a whole range of impact studies, subject to additional evaluation for the purposes in mind. While not validated specifically for floods, the encouraging evaluation outputs for higher flow percentiles suggests users can analyse high flow metrics and variability (e.g. frequency of flows above a threshold), even if not annual maximum peak flows.

As with FFGWL, there are a number of advantages of using eFLaG for future projections: it is a spatially coherent dataset, meaning that future changes in hydrological variables can be compared between catchments, boreholes and aquifers at the regional-to-national scale. This is a key benefit for both research as well as practical water resources planning. Spatially coherent projections are needed to address the spatio-temporal dynamics of droughts (e.g. Tanguy et al. 2021) and how these may change in future and what this may mean for water resources planning – where, in practice, water resources management plans often involve transfers between regions (e.g. Murgatroyd et al. 2021). Tanguy et al. (submitted) address the changing future spatial coherence of droughts using eFLaG.

Another key benefit of eFLaG is that transient time series (daily data from 1980 to 2080) allow users to can explore the future evolution of river flow and groundwater variability on interannual and decadal timescales, rather than just using 'Change Factor' approaches that compare between future time slices and the baseline.

The use of an ensemble of outputs enables users to consider uncertainty in driving data (via the 12 member RCM ensemble) as well as, for river flows, hydrological model uncertainty. In addition, different models provide different benefits: G2G performs less well against observations than the (calibrated) lumped catchment models, but does enable the characterisation of natural flows, which is vital for some uses (e.g. in providing naturalised river flows for regionalisation or as a baseline for assessing impacts, as common in regulatory and hydroecology applications e.g. Terrier et al. 2021). Moreover, abstractions and discharges can be added to the naturalised runs, as demonstrated by Rameshwaran et al. 2022. This opens up the possibility of projecting the evolution of future naturalised and impacted river flows separately – a follow-up study on this topic is underway by the authors.

Furthermore, G2G's response to rainfall may be less tailored to the present-day climate than
the calibrated models, as noted in the limitations section. The eFLaG hydrological model
ensemble therefore includes models that may be beneficial for different applications
according to the particular needs of end-users.

**Limitations and guidance**

Users of the eFLaG dataset should be aware of its limitations. While the evaluation shows
encouraging results at the national scale, there are inevitably some catchments and
boreholes where the evaluation (either Stage 1, Stage 2 or both) indicates poorer quality
simulations. Users must be aware of this, and should consult all the provided evaluation
metrics when considering which catchments to use (and which models to use) in their
analyses.
Users must also be aware that while there is some consideration of uncertainty through the
adoption of the RCM PPE, and the use of a multiple models for river flows, there are many
other sources of uncertainty not sampled in eFLaG. While the PPE gives a range of 12
outcomes, it is only one UKCP18 product and one emissions scenario, so does not sample
the full range of outcomes in UKCP18. The emissions scenario, RCP8.5, is considered to be
a pessimistic scenario (Hausfather & Peters, 2020), so this should be borne in mind, and the
eFLaG projections (along with other uses of the UKCP18 Regional projections) can arguably
be seen as akin to a 'worst case' for planning (Arnell et al. 2021).  Future work should position
eFLaG against the wider range of UKCP18 outcomes.
Furthermore, only one bias correction approach is used. Although we use a range of river
flow models, clearly other hydrological models could provide different outcomes than the set
used here, and we have only used one groundwater level model and recharge model
respectively so have not considered model uncertainty for groundwater. We have also not
considered other sources of uncertainty in the hydrological modelling (e.g. parametric
uncertainty, as in e.g. Smith et al. 2019), nor the impacts of different observational driving
climate datasets (e.g. different formulations of Potential Evapotranspiration, as in e.g. Tanguy
et al. 2018). These studies demonstrate these can be significant sources of uncertainty, but
it was beyond scope to consider within the resources available to eFLaG given the high
number of existing runs – future studies should address this.
The eFLaG modelling framework adopted the approach of calibrating using a full period-of-
record, rather than using a split sample approach. Given the length of record, this is unlikely
to be too significant (as shown for GR4J in the UK by Harrigan et al. 2018) relative to using
split sampling, but at the same time, uncertainties inevitably remain about future projections
well outside the calibration period, not least given likely non-stationarities in catchment
properties. It should also be born in mind that strong performance of a model as indicated by
good metric values is not necessarily a reliable indicator of a model's ability to reproduce

trends in hydrological signatures such as those describing low flows (Todorović et al. 2022) – this is particularly the case for the future under a changing climate.

Following on from this, one important limitation of this study – in common with the original Future Flows product (Prudhomme et al. 2012), and indeed a great majority of climate projections in hydrology – is the lack of explicit modelling of human disturbances. This is simply unavoidable as large-scale datasets of artificial influences have only recently been made available in the UK, and only for England (e.g. Rameshwaran et al. 2022). This especially applies for the lumped catchment models and groundwater level model. As such processes are not represented, they will simply be accounted for implicitly during calibration. Of course, this is unrealistic as artificial influences are likely to change in future and such non-stationarity could be locally significant. However, it should be borne in mind that the purpose of eFLaG is to model future river flow characteristics based on current catchment conditions, rather than truly chart future river flow trajectories in these catchments. For most practical applications, assuming current artificial influences and projecting forwards in time is entirely reasonable, especially in the absence of any informed understanding of how artificial influences will change.

There are also considerations for end users when applying the projections directly in impact assessments. Notably, the HadREM3-GA705 climate model that underpins the UKCP18 RCM outputs is run on a 360-day calendar year. The eFLaG projections do not modify this calendar when producing the meteorological, hydrological and hydrogeological variables and it is therefore the responsibility of the end user to deal with this in an appropriate way. There are a number of ways of doing this (e.g. Prudhomme et al. 2012; Dobor et al. 2015) and in general, there is no agreed optimal approach. Where this is performed as a post-processing step by the user (as with the eFLaG datasets), it is likely that the best approach will depend on the impact or systems modelling being undertaken.

Finally, eFLaG only provides projections for a subset of the UK gauging station network (200 catchments from some 1200 on the NRFA). This is an inevitable constraint, as with the original FFGWL product (300 locations). While we have tried to sample UK hydrology to give users as much scope as possible, there will still be a need to transpose projections to sites of interest for some users. One of the benefits of eFLaG is that gridded river flow and recharge models are used. While these gridded datasets are not yet openly available, current follow-up initiatives are looking to exploit them for providing projections at ungauged locations. A gridded dataset using G2G, but with different driving data, is described by Kay et al. 2023.

## 9. Data Availability

The eFLag dataset is associated with a Digital Object Identifier. This must be referenced fully for every use of the eFLag data as: https://doi.org/10.5285/1bb90673-ad37-4679-90b9-0126109639a9

All eFLaG files are available through the UKCEH Environmental Informatics Data Centre (EIDC): https://catalogue.ceh.ac.uk/documents/1bb90673-ad37-4679-90b9-0126109639a9

The data are stored as .csv files in the folder structure shown in the Guidance note available at Hannaford et al. (2022). In total there are 3304 files: one for each variable, model and catchment/borehole combination. They can be broadly split into two groups of files (Table 4), simobs and simrcm, as follows.

simobs

For the meteorological data, the simobs files contain date-indexed, observation-driven simulations (sim) data for precipitation with snowmelt and potential evaporation. For river flows and groundwater levels the simobs files contain date-indexed, observation-driven simulations (sim) and associated observations (obs) if they exist.

simrcm

For the meteorological data, the simrcm files contain date-indexed, RCM-driven simulations for the twelve RCMs used in eFLaG for both precipitation with snowmelt and potential evaporation. For river flows and groundwater levels the simrcm files contain date-indexed, RCM-driven simulations for the twelve RCMs used in eFLaG.

**Table 4.** eFLaG dataset structure information

| | Data | Name of file | Years available |
|---|---|---|---|
| simobs | Daily meteorology (precipwsnow (mm d⁻¹) + PET (mm d⁻¹)) | *ukcp18_simobs_[nrfa-station-number/borehole-name].csv* | Jan 1961 – Dec 2018 |
| | Daily river flow (m³s⁻¹) | *modelname_simobs_nrfa-station-number.csv* | Jan 1963 – Dec 2018 |
| | Daily groundwater levels (m AOD) | *AquiMod_simobs_borehole-name.csv* | Jan 1962 – Dec 2018 |
| | Daily groundwater recharge (mm d⁻¹) | *zoodrm_simobs_groundwater-body-name.csv* | Jan 1962 – Dec 2018 |
| simrcm | Daily meteorology (precipwsnow (mm d⁻¹) + PE mm d⁻¹) | *ukcp18_simobs_nrfa-station-number.csv* | Dec 1980 – Nov 2080 |
| | Daily river flow (m³s⁻¹) | *modelname _simrcm_nrfa-station-number.csv* | Dec 1982 – Nov 2080 |
| | Daily groundwater levels (m AOD) | *AquiMod_simrcm_borehole-name.csv* | Jan 1982 – Nov 2080 |

| | Daily groundwater recharge (mm d$^{-1}$) | *zoodrm_simrcm_groundwater-body-name.csv* | Jan 1981 – Nov 2080 |
|---|---|---|---|


where *modelname* is G2G, PDM, GR4J, GR6J. NRFA station numbers and borehole names are given
in the eFLaG_Station_Metadata.xlsx workbook.
The gridded bias corrected precipitation data is also made available as a separate dataset on the
EIDC (Lane and Kay, 2022): https://doi.org/10.5285/755e0369-f8db-4550-aabe-3f9c9fbcb93d

**Conditions of Use**
The eFLaG dataset is available under a licensing condition agreement. For non-commercial
use, the products are available free of charge. For commercial use, the data might be made
available conditioned to a fee to be agreed with UKCEH and NERC BGS licensing teams,
owners of the IPR of the datasets and products.

**Acknowledgments**
This study was funded by the Met Office-led component of the Strategic Priorities Fund
Climate Resilience programme (**https://www.ukclimateresilience.org**) under contract
P107493 (CR19_4 UK Climate Resilience). The authors thank the Met Office SPF team
(notably Jason Lowe, Zorica Jones and Mark Harrison) for direction, and all the participants
from the UK regulators and water industry for providing inputs to stakeholder engagement
events that helped shape eFLaG. JM, MM, MA and CJ publish with the permission of the
Executive Director, British Geological Survey (UKRI).

**Author Contributions**
JH led the study and the river flow components, JM led the groundwater level and
groundwater recharge components. AK and RL created the bias-corrected climate input data.
Site selection was carried out by SP, TC and JM. Hydrological simulations were run by KS
and TC (GR models), AR, AK and VB (G2G model) and JW, RM, SC and SW (PDM). JM and
MM produced the groundwater level and groundwater recharge simulations. SP and TC led
on evaluation and flow regime/drought analysis. CC, MD, MS, AW carried out the
demonstrator work and water industry engagement that helped design and shape eFLaG. ST
led on data management and portal development. JH led the preparation of the manuscript
with input from all authors. All authors contributed to the direction of the study and delivery of
the dataset.

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
