# Peer review of "The eFLaG dataset: development and evaluation of nationally"

_Earth System Science Data, 2022_

## Author Response (AR1)

Dear Editor

Firstly, we thank the reviewers for these detailed comments that have helped improve the clarity of the paper. We have addressed them as described below, and we hope this has improved the paper for the ESS readership.

As noted in the correspondence with our assigned editor, Lukas Gudmundsson, we have been made aware of a minor error in one of the input datasets used by eFLaG. This was very much 'upstream' of the eFLaG modelling chain and beyond our control.

However, given obvious potential relevance, we have undertaken a rigorous analysis to show that the error does not impact significantly on the eFLaG projections (of river flow or groundwater) nor on any conclusions emerging from using eFLaG to look at future drought or low flows changes.

We have prepared a technical note (eFLaG RCM PE Experiments FINAL for ESSD.pdf, included in the supplementary zip) to show this analysis, and will be sending this to the EIDC for archiving alongside eFLaG.

For completeness, we present it here so that it can be reviewed at the editors discretion, and if necessary could be appended to the ESS paper as part of the supplementary information (or some other mechanism). We welcome the views of the editorial team on this issue.

Best wishes

Jamie Hannaford and the eFLaG Team

=============================================================

**Reviewer 1**

The title of the paper is misleading, it had me expecting to read about projected changes in flows and groundwater in the UK, when in fact the paper presents an evaluation of hydrological models prior to being forced with future runs. Indeed the opening sentence of the abstract doubles down on this - the paper presents a dataset of natioanlly consistenty hydrological projections for the UK based on the latest UK Climate Projections. The paper doesnt do this and thus needs to be reframed as a paper evaluating hydrological models and RCM ensemble members in capturing observed regime - PRIOR to providing climate change projections for the sector.

>>>We feel this is not entirely correct as we *do* provide a nationally consistent dataset of river flow projections, this is exactly what eFLaG is, and as is made clear from the open dataset. The paper is a data paper and so of course much of the material is about the development of the data and this includes the evaluation (forced with past data as well as with the RCM data used for future runs). We do not feature projections themselves as that is the topic of a separate scientific paper (though note later replies to reviewer 2 that we will feature some lmited exemplare results from the eFLaG portal, a viewer that has been released subsequent to this paper). We modified the title to ' The eFLaG dataset: developing nationally consistent projections of future flows and groundwater based on

UKCP18'. We must still emphasise that this is about a dataset of hydrological projections.

Apart from this major oversight, the paper is well written and presented. There are a number of questions I was left with after reading that should be addressed in a revision. First, why the differential sampling of models for flows and groundwater? 3 different hydro models but only one GL model and one recharge model. In addition, why was uncertainty in model parameters not investigated.

>>>Thank you for the positive assessment of the paper's communication. The different choice of models reflect availability and suitability of models for both flow and groundwater. Parametric uncertainty was beyond scope, give the high number of existing model runs, and was highlighted as a priority in the future look (sect. 8) – we have now stated clearly why this was not possible and reaffirmed the need for follow-up work.

Second, was any testing done on the transferability of models to future conditions? In particular the lumped conceptual models (GR and PDM) are held to perform better for modified catchments because the calibration picks this up. However, for future runs for later in the century are these modifcations likely to be stationary and therefore how representative are these models for these catchments likely to be? On a related note, it would be useful to descibe the range of distribubances affecting catchments - is it predominatly abstractions?

>>> This is always the perennial problem with future runs using calibrated models. However, the purpose of eFLaG is not really to look at such questions of non-stationarity in catchments, abstractions etc, which is a major topic in itself. Rather eFLaG is about running the models under current conditions and exploring future climate impacts. We added some caveats to the discussion to emphasise this point explicitly. Seperately, a follow-up study is already looking at future changes in abstractions/discharges as well as natural flows. We highlighted this avenue for future work in section 8. The range of disturbances are broad and users can explore these through the NRFA 'Factors Affecting Runoff' codes, and we highlighted this. Abstractions are prominent, but in some catchments discharges entirely balance or outweigh abstractions. We have not highlighted this as it is tangential, in our view, but pointed the reader to several papers that highlight the extent of abstractions/discharges and their impacts.

IF the purpose of the dataset is to inform adaptation planning why was RCP8.5 selected and why not more scenarios. There is debate in scientific literature presently about the realism of RCP8.5 - its selection here needs to be justified further. I understand the need for crystalising the uncertainty as stated, but this needs to be done based on the objectives or intended uses of the dataset.

>>> This is a fact of the climate projections – the regional projections are chosen as they are spatially coherent and transient but there is only one scenario. This is a limitation of that product. We did highlight this in the caveats, but have strengthened it and highlighted the need to look at other projection.

Am I correct to conclude that there is no evaluation of model performances for an independent verification period, ie. that all metrics presented are for calibration period? How then can we be sure that calibrated models perform outside of the period used to

train them, even during the baseline period, never mind under changed future climates. Please justify NOT using a verficiation period and what this may mean for your results.

>>>We could in theory have split the observed flows into a calibration and unseen validation period (used for calculating the statistics only) and then later recalibrated on the entire period to get the best calibration for use with RCMs. But, given we typically have many years in our calibration period, this would not make much difference. The paper by Harrigan et al (2018) using GR4J (see table 2 especially) shows this. We have added further text to the 'limitations' section to discuss this too.

Minor comments

Some aspects of the uncertainties explored seem to be missing from Table 1 - eg. The first sentence of section 3 mentions 2 regional climate models, groundwater models are also not included.

>> this is just the wording, what is in the brackets is the GCM and RCM separately, we reworded to avoid ambiguity. We added groundwater to the table.

Line 196 can you say how this was done for clarity

>>not sure what is not clear, here we are saying what we are not doing (infilling to get to 365 like the Prudhomme 2012 study). We have not changed this.

Line 216 - it would be useful to provide a brief overview of this downscaling method here. It is an important part of the methods and this is the go to paper descibing the dataset.

>>>we omitted this for brevity since it is a standard method used in many previous studies and well described in these references. In response to this comment we added the following: 'The bias-corrected precipitation products were then downscaled from 12km to 1km based on the distribution of the observed Standard-period Average Annual Rainfall (SAAR) over the period 1961-1990, as in previous studies (Bell et al., 2007; Kay & Crooks, 2014). This involved calculating the ratio of the observed SAAR at 1km to the observed SAAR averaged over 12km, and then multiplying RCM values by this ratio. This ensured that the spatial variability of rainfall was captured, but the total rainfall across the original 12km RCM grid cell remained unchanged.

Line 236 - what does copied down mean here - lacks clarity.

>>Agreed, we now describe more clearly, adding 'by simply setting all 1km grid cells to the value of the containing 12km grid cell.'

Figure 2 - there are 12 PPE members from text but 15 columns. Not sure I follow, is there an ensemble mean presented?

>>there are 12 columns, but the labelling is not 01 – 12 but 01 – 15, with some missing (e.g. 14). This is just the nomenclature of the RCM runs. We now make this clear in the caption and also in the main text.

I like how there is tracability between FFWGL and this dataset in terms of models, catchments etc. This is sensible and useful. I also like how research and industry needs fed into catchment selection. I also like the description of stage 2 assessment and its expectations. Dont often see this as nicely explained.

>>Thank you very much!

Line 604 - this concerns me and needs teasing out a bit more, especially given the objective of informing adaptation in the sector. What does to a degree mean? I can understand the need to assess future changes relative to the 'real' present (ie disturbed) for planning but what are the implications and cautions that should be borne in mind?

>>>We agree that 'to a degree' is ambiguous and warrants fuller explanation. As highlighted above, that is exactly our objective, climate changes relative to the real (disturbed) present – as with the original, widely used FFGWL product, which also did not take account of human influences.

We added further clarification in the discussion, around artificial influences to highlight that this is an obvious simplification – entirely justifiable for most planning purposes, but one which needs to be considered in other applications where users may be interested in future changes in such influences. As noted, having G2G is an advantage as it does allow one naturalised run for comparisons.

Figure 4 caption needs to state #catchments and refer to table detailing the skill scores.

>>>edited

Line 644 missing word.

>>>edited

 Line 680 full stop missing.

>>>edited

Line 840 - should really discuss the transferability of models and parameter sets from current to future conditions as a key source of uncertainty here, along with the stationarity of disturbances in relative catchments.

>>>agreed, as above, we have highlighted this in the limitations

**Reviewer 2**

This manuscript deals with a dataset of multimodel hydrological projections across the UK. It presents in a detailed way how this dataset has been produced, but it lacks a description of the dataset itself. I would therefore recommend a major revision for the

authors to bring in an overview of what's actually in this (otherwise important) dataset. I would therefore also appreciate more detailed comments on how to use it, notably for interpreting hydrological projections where models have been calibrated against anthropogenically disturbed streamflow observations.

>>> We describe the dataset in section 9. For how to use the data, this is the subject of a series of separate demonstrators we describe briefly, but which are being written up elsewhere. For this data paper we cannot highlight all use cases, and we have already highlighted some of the uses of the original FFGWL data. We agree have amended the 'applications' and 'limitations' sections accordingly to reflect on the issue of artificial influences, highlighting uses of the data and appropriate caveats. We have clarified the issue with anthropogenic disturbances, as highlighted by reviewer 1 too, and modified Section 8 accordingly.

**Main comments**

- *Calibration period*: Despite (too) long descriptions of hydrological model set-ups, there are important missing information on the calibration of conceptual models. First, I haven't seen the calibration period explicitly mentioned in the text. I assume that is the complete historical 1961-2018 period covered by simobs simulations, but this is to be made explicit.

  >> also noted in response to R1, we have clarified this

- *Parameter transferability*: the previous comment calls for another question on the transferability of parameters of conceptual models, which has been shown for some time as an important issue when dealing with climate change (see e.g. Thirel et al., 2015). This issue is unfortunately completely absent from the manuscript, even in the discussion part. This issue is often dealt with by using more or less advanced split-sample set-ups, but many alternative propositions have been made over the recent years (e.g. Todorović et al., 2022). This issue should therefore be dealt with in the manuscript, at the very least as a comment in the discussion part.

  >>> we will clarify in the method and also in the discussion, and will add the reference highlighted as good examples, thank you.

- *Calibration on catchments with anthropogenic disturbances*: This issue is seemingly considered as a rather light one in the manuscript (L604-615). I rather disagree here, as the main underlying hypothesis is not even mentioned here: models calibrated on influenced data will deliver hydrological projections of influenced streamflow in which the amount and seasonality of anthropogenic disturbances (abstractions, reservoir management and so on) is equal to those during the calibration period. Which is clearly an unrealistic feature of the future. This makes the understanding and the use of such projections quite difficult for water managers and stakeholders. I hope that communication around the EFLaG dataset can handle this issue, but it is definitely not convincing in the manuscript.

  >>> we have addressed this in the method and also in the discussion. Of course, we recognise that disturbances will change in future. But it is not really our intention to accurately model catchment flows to the 2080s – rather, to look at potential climate futures for each catchment given current conditions, for planning purposes, and as widely adopted in many other appplications (including the original FFGWL, the model for

eFLaG). We have modified this accordingly in the modelling section and the discussion, as also noted in our replies to R1.

- *Description of dataset*: as already pointed out by Anonymous Referee #1, this manuscript describes how the eFLaG dataset has been built (which is definitely commendable), but it does not describe the dataset itself. Indeed, there is no e.g. (1) overall summary statistics (temporal or spatial) of present-day-period simulated streamflows or groundwater levels, (2) overall summary statistics of projected evolution or changes over the UK, and (3) no case study example of the numerous time series produced. All these features are essential to get a grasp of what's in the dataset and are therefore in my view a required feature of a data paper.

>>> We disagree, in general, on this being a required feature as we have described the provenance of the data in detail – how it was made, and evaluated, rather than summary statistics. Summary statistics and case studies are part of extensive follow-up publications, two of which are nearing submission. The results (as well as the evaluation) can also be explored on the eFLaG Portal (https://eip.ceh.ac.uk/hydrology/eflag) which, now published, we refer to more fully in the revision. We added a section in a new 'uses of eFLaG' section, and include a few plots from the portal to 1) demonstrate what can be seen on the portal, and 2) show some simple changes to low flows, which will also help address comments from Reviewer 1.

- *Length of model descriptions*: I guess that the manuscript is currently quite lengthy because of the extent to which hydrological model set-ups are described, at the expense of the more general and important issues listed above. A new balance should be reached in the revised version of the manuscript.

>>>We feel it is more important for users of the data paper to understand the provenance (i.e. methodology) rather than the outcomes, which are the subject of multiple papers and reports.

**Specific comments**

- Table 1: I disagree with the partitioning of uncertainties here: "model structure" or "model choice" (or here "hydrological models" like "climate models") are equivalent. GR4J and GR6J are indeed different models, probably a bit similar to each other than to PDM for example, but "model structure uncertainty" is commonly used as opposed to "model parameter uncertainty" in common uncertainty decomposition of hydrological projections (see e.g. Christierson et al., 2012 for the UK). This relates to one of my main comment on parameter temporal transferability.

>We are following the logic and nomenclature of the Smith et al. 2018 paper here. We see GR46J and GR6J as two structures. We added a line on parametric uncertainty but note we do not sample it (see replies to R1) – this was on an earlier version but omitted.

- L178-189: Please recall a reference for UKCP18

>>will do, we refer to Murphy et al. at line 55 but reiterate here **[complete]**

- L190-196: This issue with Hadley Centre models has been around for the last 15 years at least… But here it means that simrcm streamflow series also have 360 days per year? This is what I can see from the data files, but this is not discussed or even mentioned in the manuscript. I wonder what a water manager would say when looking at those files… It would therefore be necessary to rise the issue in the manuscript and also provide some advice for water managers and stakeholders on how to use such unusual time series, in order to prevent any misuse of even rejection based on lack of credibility.

>>>We have highlighted this in the text, and also in the discussion section (as noted with R1 replies, we added a short 'applications' section). The issue of 360 day years is also highlighted in the demonstrators, with appropriate recommendations.

- L215-217: This spatial disaggregation step is not clear enough. Plus, what is the standard-period in SAAR? And why is HadUK-Grid not used here? All these choices are not enough commented and justified.

>> This is a very standard approach but added extra text to clarify this, as detailed in replies to R1. SAAR is used as a standard, not subject to change as HadUK is.

- L228-236: Could you give here a simple description of the PET formula (e.g. "Penman-Monteith-like")? Indeed, the choice of PET formula may have strong consequences especially for low-flow changes (see e.g. Lemaitre-Basset et al., 2022). This choice would also deserve a comment in the manuscript.

>> This follows the CHESS Method as highlighted with a reference so we did not expand, but we will add a short additional sentence to make clear. We agree PE formulation can be important and we highlight this already in Section 8, L849. Thanks for the additional reference.

- Figure 2: The bias-correction factors are quite high for some month/model. This should also deserve a comment, especially with the somewhat overlapping issues of model weighting versus internal variability.

>>> we have added a comment about the high bias correction factors

L291-294: In relation to one of my main comments, I could not find in the eFLaG_Station_Metadata.xlsx file any flag indicating a near-natural catchment (e.g. belonging to UKBN2) or borehole that would help identifying locations where streamflow/groundwater projections are natural streamflow/groundawater projections. This lack of flag (I noted the FARL field, but this is far from being the only relevant source of disturbances) makes me uncomfortable with this definitely rather non-homogeneous dataset.

>>>We do highlight UKBN2 membership in the spreadsheet (Column I). We do not refer to other sources such as NRFA descriptions or FAR (Factors affecting runofff) codes, but we will add a sentence to note these sources are available.

- Figure 3: The text is very small and makes maps difficult to read.

>> we would hope this would be made a large map in the paper. We cannot really make the text much bigger without making it too cluttered.

- L395: The CHESS version cited here has been superseded.

>>we have highlighted this

- Table 3: It is necessary to have this table in the main text?

>>> yes, it is necessary to allow the reader to look at the metrics close to where they are cited. This would be our preference.

- L555-559: How is SGI used for evaluation? This is unclear until Figure 5 a few pages later when we learn about the NSE_SGI.

>>>SGI is used as the basis for comparison of the observed and simulated data, with the various metrics used to establish performance. We have made this clearer.

- L634: already written above.

>>not sure what is meant here, there is a general point made and then a specific one wrt Rudd et al. 2017.

- Figure 5, caption: "NSE_SGI"

>>agreed we have made the caption and text consistent.

- Figure 7: This figure is definitely too small for it to be correctly read and interpreted (see e.g. L698-700). What about using instead a metrics (or a very few metrics) based on the correspondence between FDCs? This would allow showing all locations in the main text.

>>we prefer to show for individual catchments. We imagine this would be a large image in the paper (see also appendices).We do in fact show metrics based on correspondence of specific FDC quantiles, e.g. Figure 9 for Q90 and then the various equivalents in the Supplementary info.

- Figure 9: This is a poor choice of color scale which makes contrasts much too difficult to read. I would highly recommend using one of the scales available at https://colorbrewer2.org and recommended by the IPCC (2018), and reserve high values for darker colors, for them to be emphasized. Plus, the legend is repeated.

>>>We agree this could be revisted, thanks for the suggestion.  We used a a palette from colorbrewer2, but to be honest, it'll still be difficult to distinguish between adjacent categories -- that's the nature of these palettes, and we feel that it does not effect the interpretation. Re: Legends, there are two legends that are almost identical but one is for Q90 (for the 4 HMs) and the other is for L90 (for g/w)

- Figure 10: Same comments as above. Impossible to distinguish between 2-5, 5-10, and 10-20 classes.

**Technical corrections**

- L105: "EdgE"
- L126: "a,b,c"
- Table 1: Please define "PPE" (Perturbed Physics Ensemble)
- L274: Please define "EIDC"
- L644: missing "models"
- L694: "re"?

>>> thanks, have made these changes

---

## Author Response (AR2)

**Reviewer 1**

The authors have done a good job at responding to points raised in the first review. In particticular they have dealt with points of clarification and added further discussion on the limitations of the dataset, especially regarding the transferability of hydro model parameters. Overall it is my conclusion that the paper is ready for publication with some further minor corrections, which i outline below. I am happy for the editor to sign off on these.

>>we thank the reviewer for this assessment of the improved manuscript

In my first review i noted that the title, abstract and intro of the paper gives the sense to the reader that the paper will present the final dataset of projected change in flows, which the paper does not actually do. Rather the paper describes the development and evaluation of the dataset while further papers are in train to describe the final data and projected changes. The authors have edited the title in this review but I still think that this does not actually say what the paper does. This could be easily addressed as follows:

Change the title to: The eFLaG dataset: development and evaluation of nationally consistent projections of future flows and groundwater based on UKCP18

>>We agree that this helps so we have changed to this

Abstract: The abstract begins with 'this paper presents the dataset'. It doesnt, it presents the development and evaluation of the methods to produce the dataset. It is an important difference. The abstract should therefore open with 'This paper details the development and evaluation of...'

>>We have changed this too

Introduction: same point as abstract, need to emphasise development and evaluation in the opening sentence.

>>As above

The biggest limitation of the dataset is use of only a single climate model and emission scenario. This is certain to underestimate the range of plausible change in future flows. While these points are dealt with in the limitations section they are important, especially given the intention of informing water management. Therefore I suggest that a sentence on key limitations is included in the abstract.

>>We have added that to the abstract

At bottom of page 6 (I am reading the tracked changes version) it is stated that model structure uncertainty is accounted for by considering two versions of one of the models. This is a minor part of this evaluation, inclusion of physically based gridded model and PDM is more structurally different that two versions of the GR models. This sentence needs to be edited to simply saying uncertainty in model structure is accounted for.

>>We considered the difference between GR/PDM and G2G to be model uncertainty (diff models) and ArGR to be structural (in the sense of different structures of same model type). But agreed there are not always clear distinctions between this. We have edited accordingly.

Page 8 when describing biases in precipitation, give some numbers in the text to help reader quantify the magnitude of biases, at present only descriptives are provided such as substantially over estimates.

>>we have added some context

The authors note that the approach taken in developing the study design vis uncertainties was to crystalise the uncertanties. Going back to the Smith et al paper crystalise is described as sampling the spectrum of potential results at each step to reduce or "crystallize" salient outcomes. The selection of a

single climate model/emissions scenario and then more heavily sampling hydrological model components of the cascade does not seem to me to match this description.

>>We agree this is technically not correct, and amended accordingly

Page 16 reduce the number of times the term described below is used, at least three times in quick succession.

>>we have changed this

Page 17 reduce the number of times the term a range of metrics is used, at least three times in quick succession

>>we have changed this

Page 22 What are increasing agressive stages?

The stages of the calibration optimise increasing numbers of model parameters in turn (which is what was meant by increasingly aggressive). The text (repeated below) has been updated to make this more explicit. Full details are included in the Supplementary info.

"Parameter estimation was performed using an automatic calibration procedure that applied a simplex optimisation scheme (Nelder and Mead, 1965) to increasing numbers of model parameters in turn."

**Reviewer 2**

80: See Chan et al for how climate projections have been used in drought and climate change assessments: https://doi.org/10.1177/03091333221079201

>>We added this ref

128: However, they have been used in industry in support of WRMP24

>> we meant in terms of the scientific literature, but we can highlight this – we referred to Thames water's WRMP. We note that these are draft and still not readily available.

328: Sites or/and catchments?

>>We use interchangeably at various points, we think this is clear

333/417/943: Artificial influences captured indirectly via calibration on observed flows (make this clearer from the beginning). Standard industry approach is to do climate change assessment on naturalised and then denaturalise using artificial influences.

>> We thought we did make this clear right at the start of the modelling section now (417), and we do ref it indirectly at 333 when talking about catchments. Hard to see how we could ref earlier without being 'too early' before the context.

515: Descriptions of models are quite long and could be moved to supplementary material

>>we disagree, there is a lot of SI and we felt the need to keep a minimum of the model description in the body. We trimmed superfluous material last time. We feel this should now stand (reviewer 1 picked this up last time but not again this time).

590: Where was the spring factor used – were there 'rules' used to identify relevant catchments?

In PDM, the spring factor is a multiplicative factor on the baseflow used to represent net losses or gains of water to permeable catchments (e.g. from external springs). Because of this we used a criteria based on the Base Flow Index (a measure of the relative proportion of base flow), BFI > 0.7, to identify potential catchments for which the use of the spring factor might be hydrologically plausible (the only exception being the Leven at Linnbrane, strongly affected by Loch Lomond, where we deemed its use inappropriate). Then, for such catchments, we chose either to use a model configuration using the spring factor, or one using the rainfall factor, according to which produced the best performance in the calibrated modelled flows. Full details are included in the Supplementary info.

983: add apostrophe in "model's"

>>> Done

984: this is particularly relevant in future with changing rainfall regimes

>>we will add a sentence to this effect.